# Joint Structure Search for Tensor Network Operators Inspired by Symmetry Breaking

## Abstract

Tensor networks (TNs) offer a compact representation for high-dimensional operators in physics and machine learning. While TN structure search (TN-SS) has advanced model selection, prior work is limited to a *single* operator. Yet real systems, such as transformers and quantum circuits, would contain multiple coupled operators, where treating them independently or enforcing a single shared structure is fundamentally limiting. We introduce *joint TN-SS*, the first framework for multi-operator structure search. Our physics-inspired algorithm runs in two phases: a symmetry phase, where standard TN-SS finds a shared structure capturing common inductive bias; and a symmetry-breaking phase, where operator-specific diversity emerges through greedy core masking, guided by task-explainable loss tolerances. Across tensor decomposition, parameter-efficient fine-tuning of LLMs, and quantum circuit optimization, joint TN-SS delivers more compact representations with equal or better accuracy than state-of-the-arts, with affordable search cost. These results demonstrate that symmetry-driven diversification offers a simple, general, and scalable solution to TN structure selection in multi-operator systems.

## 1 Introduction

Linear operators are the foundation of modern computation. In machine learning (ML), linear operators form the backbone of fully connected, convolutional, and attention layers, but their massive dimensionality makes them computationally and memory intensive (Desislavov et al., 2023). Tensor networks (TNs) offer a principled way to restructure high-dimensional Hilbert space into compact, scalable representations (Oseledets, 2011; Orús, 2014; Memmel et al., 2024). There is consequently increasing interest in the efficient representation of linear operators via TNs across communities of ML (Novikov et al., 2015; Stoudenmire & Schwab, 2016; Richter et al., 2021; Yang et al., 2024; George et al., 2024; Saiapin & Batselier, 2025), statistics, computational physics, and beyond.

The growing application of TNs raises the central challenge of *tensor network structure search (TN-SS)*: how to find the optimal TNs' models (known as *TN structures*), involving not only ranks (Kodryan et al., 2023; Zheng et al., 2024), but also topologies (Li & Sun, 2020), and permutations (Li et al., 2022; Zeng et al., 2024a), for a specific task. Because TN-SS is known to be NP-hard (Hillar & Lim, 2013) and highly combinatorial, prior methods focus almost *exclusively* on optimizing the structure for a *single* TN. Finding jointly the optimal structures for *multiple* TNs within a system is considered impossible, as computationally intractable due to the combinatorial explosion.

However, heterogeneous TN structures are an empirical necessity in a learning system. Numerous studies have shown that the structural complexity of linear operators in deep neural networks varies substantially across layers. For instance, transformer weights often exhibit low-rank structure, but the low-rankness differs dramatically between layers (Wang et al., 2024; Jaiswal et al., 2024; 2025). Similar findings in adaptive pruning and low-rank adaptation further highlight that different operators respond unequally to factorization or sparsity (Frantar & Alistarh, 2023; Yang et al., 2024). These results expose a key weakness of existing TN-SS methods: *they ignore inter-layer heterogeneity by forcing either isolated optimization or rigid sharing across operators*. This motivates our thinking of how to optimize heterogeneous yet efficient TN structures across multiple operators without incurring a combinatorial explosion in search cost.

To address this challenge, we introduce for *the first time* a formal formulation of joint TN-SS, together with a simple yet efficient algorithm inspired by the principle of symmetry breaking in

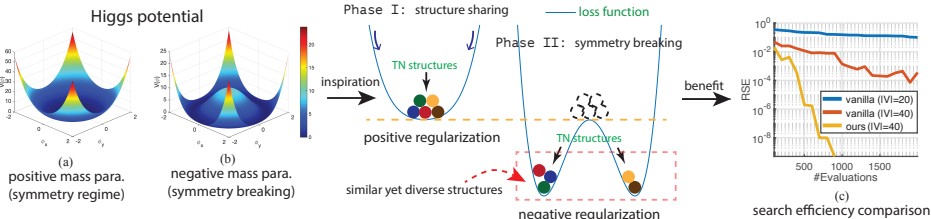

Figure 1: **Symmetry breaking (SB) in joint TN-SS.** Subfigures (a-b) illustrate the SB in physics (cms, 2022) and the relationship to the proposed algorithm. Subfigure (c) demonstrates the search efficiency of the proposed methods compared to *vanilla* TN-SS methods; experimental details are provided in Section 4.1.

physics (Lee, 1974). See Figure 1 for illustration. Just as symmetry breaking lowers energy in physical systems, as shown in Subfigure (a-b), we notice, in the context of joint TN-SS, a shared structure (i.e., a symmetric solution space) *always* exists across operators but is typically suboptimal. More efficient solutions emerge when introducing controlled diversity *close to* that shared structure. Guided by this principle, we develop a two-phase algorithm: `Phase I` enforces structural sharing for search efficiency, and `Phase II` introduces operator-specific diversity through greedy-like core masking. A unified loss (5) governs the transition, with task-specific tolerances replacing opaque hyperparameters for practical explainability. Extensive numerical results across domains, including joint tensor decomposition, parameter-efficient fine-tuning of LLMs, and quantum circuit decomposition, confirm that our approach consistently produces more compact and effective TN representations than the existing TN-SS methods with minimal overhead. Our contributions are summarized as follows:

- We introduce the framework for joint TN-SS, extending TN-SS *for the first time* to multi-operator systems and addressing heterogeneity ignored by prior work;
- We propose a simple yet efficient algorithm for joint TN-SS that balances efficiency with expressiveness through structured diversification;
- We demonstrate broad applicability and consistent gains in parameter efficiency, scalability, and accuracy across domains.

## 1.1 RELATED WORKS

**Tensor networks in machine learning.** TNs generalize classical matrix and tensor decompositions, and have become a standard tool for representing vectors and operators in extremely high-dimensional spaces (Kolda & Bader, 2009; Cichocki et al., 2017). The TN "toolbox" contains a rich collection of scalable models, including tensor train (also known as MPS/MPO) (Oseledets, 2011), tensor ring (TR) (Zhao et al., 2016), and tubal-SVD (Kilmer & Martin, 2011), etc.. In machine learning, TNs have been applied across a broad range of tasks, including model compression (Novikov et al., 2015; Kossaifi et al., 2020), multi-model learning (Hou et al., 2019), efficient training and fine-tuning for large language models (Yang et al., 2024; Chen et al., 2024a; Tao et al., 2025), reinforcement learning (Sozykin et al., 2022), prompt learning (Qiu et al., 2025), and statistical learning (Stoudenmire & Schwab, 2016; Han et al., 2022; Saiapin & Batselier, 2025). In these studies, the most recent studies (Chen et al., 2024a; Li et al., 2025) show that TNs with topology inspired by quantum circuits, which we call in this paper *tensor network operators* (TNOs) (Hackbusch & Kühn, 2009; Bensa & Žnidarič, 2021; Rudolph et al., 2023) can offer substantial advantages in both performance and computational efficiency. Motivated by these new discoveries, our work focuses on optimizing TN structures specifically in circuit-based formats following this line.

**Tensor network structure search (TN-SS).** TN-SS extends classical TN rank selection (Babacan et al., 2012; Rai et al., 2014; Zhao et al., 2015; Yokota et al., 2016) to richer structure-related hyperparameters such as topology and tensor permutations (Cheng et al., 2020; Mickelin & Karaman, 2020; Li et al., 2021a; Kodryan et al., 2023; Hayashi et al., 2019; Hashemizadeh et al., 2020; Li & Sun, 2020; Haberstich et al., 2023; Chen et al., 2024b; Zheng et al., 2024; Li et al., 2023; Zeng et al., 2024a; Guo et al., 2025). Despite steady progress, two major gaps remain. First, the structural modeling of those circuit-like TNs has never been systematically formulated. Second, existing TN-SS methods almost exclusively target a single TN or enforce one shared structure across operators,

ignoring inter-operator heterogeneity. This work closes both gaps. We provide the first formal formulation for the structural representation of circuit-like TNs and introduce joint TN-SS, the first framework for simultaneous structure search across multiple operators within a unified system.

## 2 BASICS OF JOINT TN-SS

We introduce the key concepts of tensor networks, present the new formulation of joint TN-SS, and close with a brief review of symmetry breaking, the central inspiration for our search algorithm. To ensure precision, we present these concepts mathematically in the main text. An accessible, intuition-oriented exposition is also offered in the Appendix.

**Notations.** We use $\mathbb{R}$, $\mathbb{C}$, and $\mathbb{Z}_{>0}$ to denote the sets of real numbers, complex numbers, and positive integers, respectively. When both fields are admissible, we use $\mathbb{F} \in \{\mathbb{R}, \mathbb{C}\}$. Bold lowercase and uppercase letters (e.g., $\mathbf{x} \in \mathbb{F}^N$, $\mathbf{A} \in \mathbb{F}^{I \times J}$) represent vectors and matrices, respectively, and calligraphic letters (e.g., $\mathcal{A}, \mathcal{B} \in \mathbb{F}^{I_1 \times \cdots \times I_N}$) denote tensors. For a vector $\mathbf{x} \in \mathbb{F}^n$, $\text{diag}(\mathbf{x}) \in \mathbb{F}^{n \times n}$ denotes the diagonal matrix whose diagonal entries are given by $\mathbf{x}$. The symbol $\otimes$ is used for the tensor product, and $\bigotimes$ denotes a sequential application of the products. We use $\times$ for the Cartesian product *between sets* and $\circ$ for function actions on graphs. The space $\mathbb{H} := \mathbb{F}^{I_1} \otimes \cdots \otimes \mathbb{F}^{I_N}$ is written as $\mathbb{F}^{I_1 \times \cdots \times I_N}$ or $\bigotimes_{n=1}^N \mathbb{F}^{I_n}$ for brevity. The notation $|\cdot|$ denotes norm-like quantities depending on context: e.g., $|\phi|$ is the absolute value for $\phi \in \mathbb{C}$, and $|G|$ denotes the order (number of vertices) of a graph $G$. Last, we define $[K] := \{1, 2, \ldots, K\}$ for convenience.

### 2.1 TENSOR NETWORK OPERATORS

**Tensor network (TN).** We adopt the formal definition of TNs by Ye & Lim (2019), where a TN, denoted by $tns_{\mathbb{H}}(G, r, p)$, is defined as a set of tensors in a space $\mathbb{H}$. Here, $G = (V, E)$ is a graph that defines the TN topology (Li & Sun, 2020), where the vertex set $V$ corresponds to a collection of core tensors, while each edge in $E$ indicates a tensor contraction between connected tensors. The function $r : E \to \mathbb{Z}_{>0}$ assigns a positive integer to each edge, specifying the TN ranks. Additionally, the mapping $p : \{\mathbb{F}^{I_n}\}_{n=1}^N \to V$ associates each input subspace $\mathbb{F}^{I_n}$ with a vertex of $G$, specifying the TN permutation (Li et al., 2022). For a more detailed introduction to TNs, we refer readers to the overview (Cichocki et al., 2016).

**Tensor network operator (TNO).** In this work, we focus on TNOs, a special class of *circuit-like* TNs designed to represent linear operators, as explored in recent studies (Li et al., 2021b; Liu et al., 2024b; Chen et al., 2024a). Formally, let $Q, K, I \in \mathbb{Z}_{>0}$ with $K \leq Q$. A TNO of order $(Q, K)$ and dimension $I$ associated with a graph $G = (V, E)$ defines a subset of TNs, denoted by $tno(G; Q, K, I) \subseteq tns_{\mathbb{H}}(G, r, p) \subseteq \mathbb{H} := \bigotimes_{n=1}^N \mathbb{F}^{I_n}$. The construction imposes three conditions: *1)* $N = 2Q$; *2)* each mode dimension satisfies $I_n = I$ for all $n \in [2Q]$ with ranks fixed $r(e) = I$ for all $e \in E$; *3)* the graph $G$ with mapping $p$ is arranged such that every core tensor admits an unfolding (Cichocki et al., 2017) isomorphic to an $I^K \times I^K$ matrix. As such, any $\mathcal{X} \in tno(G; Q, K, I)$ admits a unified unfolding into an $I^Q \times I^Q$ matrix, making the operator structure explicit.

TNOs thus compactly represent high-dimensional operators of size $I^Q \times I^Q$ through the contraction of a sequence of lower-dimensional operators of size $I^K \times I^K$. Because all core tensors are constrained to be isomorphic to square matrices, *circuit graphs* (see right figures for example) offer a more intuitive visualization of complex TNOs than traditional TN diagrams (Li & Sun, 2020; Li et al., 2022; 2023). As illustrated on the right, subfigures (a) and (b) show a

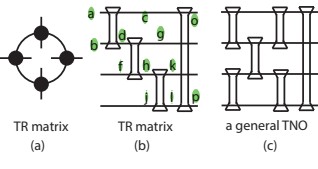

TR matrix (a)  TR matrix (b)  a general TNO (c)

TR matrix in both traditional and circuit representations, while subfigure (c) depicts a more general TNO model. In Appendix A, we further discuss the relationship and differences between TNOs and general TNs. Although TNOs may appear complex, their practical implementation is straightforward using einsum. For example, the TNO illustrated in subfigure (b) can be constructed as:

```
W = torch.einsum("abdc,dfgh,hjkl,clop -> abfjogkp", V_1, V_2, V_3, V_4)
```

Here, V_1 through V_4 correspond to the four core tensors shown from left to right in the figure. The einsum example for subfigure (c) is also provided in Appendix A.

**TNO system.** Multiple TNOs build up a TNO system. Given $Q, K, I$, we formulate a $M$-dimensional TNO system as follows:

$$ts(G_1, G_2, \ldots, G_M; Q, K, I) := tno(G_1) \times tno(G_2) \times \cdots \times tno(G_M), \tag{1}$$

where $tno(G_m) := tno(G_m; Q, K, I)$ for all $m$. This construction creates a unified combinatorial space of operators that share the same parameters $(Q, K, I)$ but may differ in topology through graphs $\{G_m\}_{m=1}^M$. We remark that each element in $ts(G_1, G_2, \ldots, G_M; Q, K, I)$ can be viewed as a vector of operators, which makes the framework able to model multiple, distinct linear operators that jointly define a system. For example, in computer vision, it can capture multi-view video tensors where each view is represented by a tensor; in deep learning, it can model a collection of linear layers in a neural network, where each layer may benefit from a different structure; and in quantum computing, it represents different unitary blocks in a circuit, which must be optimized jointly but are not identical. This perspective highlights that a *TNO system encodes heterogeneity across operators in a principled way*, going beyond the treatment of one TN in isolation.

## 2.2 JOINT TN-SS: EXTENDING TN-SS TO TNO SYSTEMS

Building on prior studies in TN-SS (Li & Sun, 2020; Li et al., 2022; 2023), we move beyond the conventional task of optimizing a *single* TN to the more general and practically relevant setting of *joint TN-SS*. Formally, given parameters $Q, K, I$, the goal is to solve the bi-level optimization problem:

$$\begin{aligned} \min_{\{G_m\}_{m=1}^M} & \min_{\mathcal{X}^M} \pi_D(\mathcal{X}^M) \\ s.t. \quad & G_m \in \mathbb{G}_m, \ m \in [M], \quad \mathcal{X}^M \in ts(G_1, G_2, \ldots, G_M; Q, K, I) \end{aligned}, \tag{2}$$

where $\mathbb{G}_m$ denotes the set of valid graphs satisfying the TNO constraints in Section 2.1, and $\pi_D(\cdot):$ $\mathbb{F}^{I^Q \times I^Q} \to \mathbb{R}$ is a task-specific loss function defined on some data $D$. Note that in the existing TN-SS literature (Li & Sun, 2020; Li et al., 2022; 2023; Zeng et al., 2024b;a) the graph $G$ is typically represented by adjacency matrices. However, such a representation makes it difficult to enforce the required TNO structural constraints. To overcome this, we introduce a *vertex-indexed incidence (VI) matrix* representation, which naturally encodes circuit-like structures. Specifically, for each TNO with graph $G_m$, we define:

$$\mathbf{I}_m = \begin{pmatrix} i_{1,1}^m & i_{1,2}^m & \cdots & i_{1,L_m}^m \\ i_{2,1}^m & i_{2,2}^m & \cdots & i_{2,L_m}^m \\ \vdots & \vdots & \ddots & \vdots \\ i_{K,1}^m & i_{K,2}^m & \cdots & i_{K,L_m}^m \end{pmatrix}, \tag{3}$$

where $L_m$ is the order of $G_m$, and the entries satisfy $i_{k,l}^m \in [Q]$ and $i_{k_1,l}^m \neq i_{k_2,l}^m$ for all $l \in [L_m]$, $k_1, k_2 \in [K]$ with $k_1 \neq k_2$. Algorithmically, the goal of joint TN-SS is therefore to optimize the set of VI matrices $\{\mathbf{I}_m\}_{m=1}^M$, such that the resulting TNO system minimizes the task loss $\pi_D$.

**Inspiration from symmetry breaking.** Our approach to joint TN-SS is inspired by the physical principle of symmetry breaking (SB), which explains how a system transitions toward a low-energy state. Unlike finding directly to the final ground state, many physical systems behave with a phase transition from a *symmetric* to a more *asymmetric* state (Lee, 1974). A classical example is by the Higgs potential (Melo, 2017):

$$V(\phi) = \kappa \phi^4 + \mu \phi^2, \tag{4}$$

visualized in Figure 1. Here, $\phi$ denotes the field magnitude, and $\kappa > 0$ and $\mu$ determine the shape of the potential. In the energy minimization process, the system first forms the potential with $\mu > 0$ (see Figure 1 (a)), converging to a symmetric state with the fastest descent. After that, the parameter $\mu$ becomes negative, the potential deforms into the "Mexican hat" (see Figure 1 (b)), and the system finally settles into asymmetric states with lower energy.

This phenomenon highlights a simple yet often overlooked principle in the study of TN-SS:

> *The optimal solution to a difficult problem often lies close to a suboptimal solution found in a coarser, lower-dimensional search space.*

In joint TN-SS, this principle guides our design. Instead of searching directly over the huge combinatorial space of all possible TNO-system structures, we begin with a symmetric structural family shared across operators. After obtaining a good coarse solution, we gradually relax this symmetry, allowing each operator to diverge from the common structure through small, targeted perturbations. This controlled symmetry breaking enables the model to discover heterogeneous, task-adapted TNO structures while keeping the overall search tractable.

# 3 THE PROPOSED APPROACH

## 3.1 REFORMULATION INSPIRED BY SYMMETRY BREAKING

For brevity, we write $ts(G_1, \ldots, G_M) := ts(G_1, \ldots, G_M; Q, K, I)$. The joint TN-SS problem (2) is reformulated by imposing a common TN structure and introducing controlled structure diversity, in direct analogy to the potential (4):

$$\min_{G \in \mathbb{G}_q, \{\epsilon_m\}_{m=1}^M} \min_{\mathcal{X}^M \in ts(G_1, G_2, \ldots, G_M)} \pi_D(\mathcal{X}^M) + \lambda \sum_{m=1}^M |\epsilon_m|, \tag{5}$$
$$s.t. \, G_m = G \circ \epsilon_m, \, \forall m \in [M].$$

Here, $\mathbb{G}_q$ denotes the set of valid graphs satisfying the TNO constraints. Each $\epsilon_m : \mathbb{G}_q \to \mathbb{G}_q$ represents a graph operation applied to $\mathbb{G}$, while $\lambda \in \mathbb{R}$ controls the strength of a regularization term penalizing the "degree" of the structure diversity. Compared to the original formulation (2), (5) assumes that all structures $\{G_m\}$ are generated from a *common graph $G$* with small perturbations $\{\epsilon_m\}$. The regularization balances the trade-off between structural sharing (when $\epsilon_m$ is close to identity) and diversity (when "big" $\epsilon_m$'s are encouraged), thereby mirroring the transition from symmetry to symmetry breaking.

It is worth emphasizing that imposing the common graph $G$ does *not* limit expressiveness. By the universal approximation property of circuit-like TNs (Barenco et al., 1995; Möttönen et al., 2004), we know there always exists a sufficiently high order $G$ that can represent all operator-specific graphs $\{G_m\}$. While highly diverse $G_m$ may in principle require a large $G$, our experiments show that in practice one can almost always find an affordable $G$ close to optimal. From this shared structure, the perturbations $\epsilon_m$ efficiently specialize into diverse $G_m$ with more compact representation for a TNO system. Hence, the common graph is not a restriction but a trick for efficiency: it reduces the initial combinatorial burden while still enabling the emergence of heterogeneous, task-adapted structures.

**Practical choice of $\epsilon_m$.** Empirically, we use *core masking* as a simple yet powerful mechanism for modeling $\epsilon_m$ in (5), offering both efficiency and effectiveness at negligible cost. As illustrated on the right, core masking corresponds to removing selected core tensors from a TNO, and the regularization term $|\epsilon_m|$ in (5) is

TNO        core masking

defined as the number of core tensors masked by $\epsilon_m$. Formally, let $\mathbf{w}_m \in \{0, 1\}^{L_m}$ be a binary mask over the $L_m$ core tensors of the $m$-th TNO of $\mathcal{X}^M$ with graph $G_m = (V_m, E_m)$ and VI matrix $\mathbf{I}_m \in [Q]^{K \times L_m}$. We have $\epsilon_m(\mathbf{w}_m) : (\mathbf{I}_m, \{\mathcal{G}_{m,\ell}\}) \mapsto (\mathbf{I}_m(\text{diag}(\mathbf{1} - \mathbf{w}_m)), \{\tilde{\mathcal{G}}_{m,\ell}\})$, where $\tilde{\mathcal{G}}_{m,\ell} = \mathcal{G}_{m,\ell}$ if $w_{m,\ell} = 0$ and $\tilde{\mathcal{G}}_{m,\ell}$ equals identity otherwise, and thus $|\epsilon_m| = \|\mathbf{w}_m\|_0 = \sum_{\ell=1}^{L_m} w_{m,\ell}$.

**Hyperparameter $\lambda$ and phase transition.** Note that the tuning parameter $\lambda$ in (5) plays the same role as the mass parameter in the Higgs potential (4), shaping the loss landscape and governing the transition between the symmetry and SB regimes. When the common graph $G$ is sufficiently expressive to have a smaller value of $\pi_D$ and $\lambda \geq 0$, the system remains in a symmetry regime: the optimal graphs $\{G_m\}_{m=1}^M$ must collapse to the single common graph $G$, since any non-trivial core masking would be penalized. In this case, each $\epsilon_m^*$ equals the identity, and the joint TN-SS problem reduces to vanilla TN-SS. This phase provides an efficient initialization by enforcing structural sharing.

When $\lambda < 0$, the system enters the SB regime. While negative regularization is unusual in ML, in physics it is precisely what drives the Mexican-hat potential to favor multiple nonzero minima. In our setting, it encourages independent, operator-specific core masking across the $\{G_m\}$, thereby breaking symmetry and introducing structural diversity. This transition expands the search space in a

---

**Algorithm 1** The proposed algorithm for joint TN-SS (sketch)

---

1: **Initialize:** parameters $Q, K, I$, tolerance $\eta_1, \eta_2$ with $\eta_1 \leq \eta_2$.
2: Define $f(\cdot) = \min_{\mathcal{X}^M \in ts(\cdot)} \pi_D(\mathcal{X}^M)$.
3: Obtain common structure $G \leftarrow$ TN-SS$(f)$ with $f(G) \leq \eta_1$  $\quad\quad\quad$ ▷ Phase I: symmetry
4: Set $G_m \leftarrow G$ with $G_m = (V_m, E_m)$ for all $m \in [M]$.
5: **repeat**  $\quad\quad\quad\quad\quad\quad\quad\quad\quad\quad\quad\quad\quad\quad$ ▷ Phase II: symmetry breaking!
6: $\quad$ Pick one core tensor $v$ from $\bigcup_{m=1}^M V_m$ at random.
7: $\quad$ Construct masked candidates $G'_m = G_m \circ \epsilon(v)$ for all $m$.  $\quad\quad$ ▷ $v$ is masked from $G_m$.
8: $\quad$ Evaluate $f' = f(\{G'_m\})$.
9: $\quad$ **if** $f' \leq \eta_2$ **then**
10: $\quad\quad$ Accept: $G_m \leftarrow G'_m$ for all $m$.
11: $\quad$ **end if**
12: **until** no further improvement is found
13: **Output:** $\{G_m\}_{m=1}^M$

---

controlled manner, guiding the system toward more compact, heterogeneous configurations that often achieve lower loss.

## 3.2 ALGORITHM

Motivated by symmetry breaking, we propose a two-phase algorithm that first enforces a shared structure for efficiency and then gradually introduces diversity via core masking.

As analyzed in Sec. 2.2, the optimal solution of (5) in the symmetry regime collapses to a common graph $G$. Thus, in Phase I we set $\epsilon_m^*$ to be identity for all $m \in [M]$ and run any TN-SS method[1] to find $G$ with $f(G) \leq \eta_1$. A small $\eta_1$ ensures $G$ is expressive, though possibly redundant. Once $G$ is fixed, the algorithm enters Phase II (the SB regime), where structural diversity is introduced by iteratively masking cores: at each step, a vertex $v$ (corresponding to a core tensor) is sampled from $\bigcup_{m=1}^M V_m$, across all TNOs, and the updated graphs are evaluated. If the task loss remains within tolerance $\eta_2$, the change is accepted; otherwise, a new masked candidate is tried. The process stops when no further beneficial masks are found. Note that the gap $\eta_2 - \eta_1$ quantifies the performance budget for pruning redundant core tensors and encouraging operator-specific diversity.

Crucially, $\lambda$ in (5) is never tuned directly in the algorithm. Its effect is implicitly encoded by $(\eta_1, \eta_2)$, which provide an explainable, task-specific rule for when symmetry should be enforced and when diversity is allowed. This yields a practical phase transition mechanism that avoids the exponential cost of unconstrained joint search. Moreover, the greedy masking ensures the number of evaluations grows only linearly with $M$ in Phase II. Experiments in Sec. 4 confirm that our method achieves more compact TN representations with little overhead.

Finally, the following perturbation bound illustrates that the loss increase from masking a core is controlled by its operator–Schmidt rank:

**Proposition 3.1** (Perturbation analysis). *Let $\mathcal{Z}^M = \{Z_1, \ldots, Z_M\}$ be a system of TNOs with $\mathcal{Z}^M \in \arg\min_{\mathcal{Y}^M \in ts(G_1, \ldots, G_M; Q, K, I)} \pi_D(\mathcal{Y}^M)$. Suppose one operator $Z_j$ contains a maskable core $\mathcal{G}$ with $\|\mathcal{G}\|_F = 1$ and operator–Schmidt rank $S$. Let $\mathcal{Z}^{M,\star}$ denote the TNO system obtained by replacing $Z_j$ with its best masked version $Y^\star \in \arg\min_{Y \in tno(G_{\mathrm{mask}}; Q, K, I)} \pi_D(\{Z_1, \ldots, Y, \ldots, Z_M\})$, where $G_{\mathrm{mask}} = G_j \circ \epsilon(\mathcal{G})$. If $\pi_D$ is Lipschitz contentious in $\|\cdot\|_F$ with respect to each operator, then*

$$\left| \pi_D(\mathcal{Z}^{M,\star}) - \pi_D(\mathcal{Z}^M) \right| = \mathcal{O}\left(\sqrt{S-1}\right), \tag{6}$$

*where the hidden constant depends only on the contracted environment of $\mathcal{G}$. In particular, if $S = 1$, masking $\mathcal{G}$ does not increase the loss.*

The proof is in the Appendix E. Prop. 3.1 shows that rank-one cores can be pruned without loss, while higher ranks incur bounded perturbations scaling with $\sqrt{S-1}$. This provides a sufficient criterion for identifying redundant core tensors, though we argue in the Appendix that low Schmidt rank is not necessary, owing to the universality of TN approximations.

---

[1] Appendix B.2 provides guidelines for adapting TNGA (Li & Sun, 2020), TNLS (Li et al., 2022), TnALE (Li et al., 2023), and Greedy (Hashemizadeh et al., 2020) to TNOs.

## 4 NUMERICAL RESULTS

We evaluate the proposed algorithm on diverse domains: joint tensor decomposition, parameter-efficient fine-tuning of LLMs, and quantum circuit optimization, to demonstrate its effectiveness. Due to space limits, we report only key settings in the main text; full experimental details are deferred to the Appendix.

### 4.1 JOINT TENSOR DECOMPOSITION ON SYNTHETIC DATA

We evaluate first the performance of our algorithm on tensor decomposition using synthetic tensors.

**Data preparation.** We choose $Q = 5$, $K = 2$, $I = 2$ and $M = 4$ for convenience, following (1), This choice corresponds to optimizing structures for tensors of order-10, which is *markedly larger* than in most prior TN-SS benchmarks, providing a challenging yet tractable testbed for evaluating our joint TN-SS algorithm. Additional results on varying $Q$, $I$, and $M$ are reported in Appendix B.5. Next, we choose four circuit-like TN structures widely used in ML or physics, including tensor train (**TT**, Oseledets 2011), "stairs-like" TN (**Stairs**, Rudolph et al. 2023), "brick-wall" TN (**Brick**, Bensa & Žnidarič 2021), and randomly-connected TN (**Rand.**), to construct the common graph $G$ in (5)[2], and then randomly mask core tensors from $G$ individually to obtain the diverse $\{G_m\}_{m=1}^M$ for each TNO in the system. Once the structures are determined, we follow the real-valued and *i.i.d.* Gaussian distribution $\mathcal{N}(0, 1)$ to generate the values of core tensors. Synthetic tensors are thus obtained by contracting the core tensors.

**Setup.** For our algorithm, we set $\pi_D(\mathcal{X}^M)$ to be the relative squared error (RSE, Li & Sun 2020) calculated globally among all TNOs, and use Adam (Kingma & Ba, 2014) to optimize the value of core tensors. The setting of $L$ is determined differently for each data, following a basic principle that $L$ should be relatively large to achieve a satisfactory approximation error in `Phase I`. The tolerances $\eta_1, \eta_2$ are set to $5 \times 10^{-10}$ and $10^{-6}$, respectively (the sensitivity analysis of $L$ and $\eta_2$ are given in Appendix B.5). We regard the approximation as successful if $RSE \leq 10^{-6}$.

**Baselines.** For comparison, we adopt four state-of-the-art TN-SS algorithms: TNGA (Li & Sun, 2020), TNLS (Li et al., 2022), TnALE (Li et al., 2023), and Greedy (Hashemizadeh et al., 2020). They also serve as the backbone optimizers for the common graph $G$ in `Phase I` of our approach. To handle TNO structures, we introduce minimal adaptations on the baselines while keeping each algorithm's original search dynamics intact. Details of these modifications are provided in Appendix B.2.

**Results.** Figure 2 (a–c) reports results across different TN-SS baselines. We first see from Figure 2 (a) that `Phase II` significantly reduces the total number of core tensors, resulting in more compact TN representations than the existing TN-SS methods. Furthermore, we see from Figure 2 (b) that such compression preserves accuracy. The approximation error remains at $RSE \leq 10^{-6}$ in nearly all cases. Third, Figure 2 (c) illustrates that the evaluation cost in `Phase II` (dark bars) is consistently lower than in `Phase I`, confirming the efficiency of symmetry breaking.

For fairness, we also tested vanilla TN-SS with the assumption of sharing structure among TNOs with the same $|V|$ as our methods, as done in many existing works. Figure 2(b) shows, this severely limits performance: none of the methods in this setting reach $RSE \leq 10^{-6}$, except TNLS on TT. This contrast highlights that structural diversity is important for achieving the full expressive power of TNOs.

**Phase transition during the search.** Figure 2 (d–f) shows the search dynamics on **Rand.** with our approach (that uses TNGA in `Phase I`). A clear phase transition emerges: once entering `Phase II` (the SB regime), the number of core tensors steadily decreases (Figure 2(d)) while RSE remains stable (Figure 2(e)). At the same time, the per-iteration evaluation cost drops sharply (Figure 2(f)), highlighting both efficiency and accuracy of the transition.

**Experimental details for cover image.** We further test a plausible baseline where vanilla TN-SS tackles joint TN-SS by composing all TN structures together into a single large graph. As shown in Figure 1(c), vanilla TN-SS (TNGA on **Rand.**, blue/red curves) is significantly less efficient than our approach ( yellow curve). This highlights the power of the shared-structure prior in (5), which guides the search toward diverse yet compact solutions at far lower cost.

---

[2]Note that our focus is on TN structures tailored for high-order tensors; thus, classical tensor decomposition models such as CP, Tucker and tSVD are excluded from our experiments.

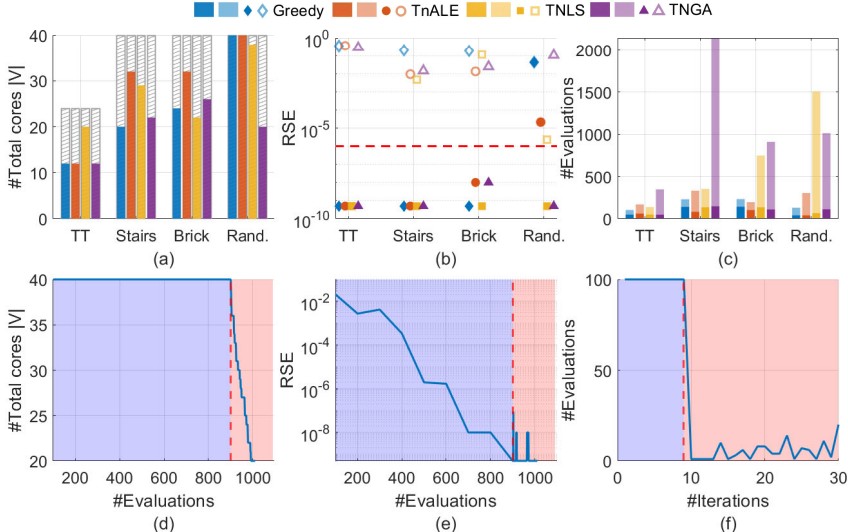

Figure 2: Experimental results of tensor decomposition. (a) Shaded and colored bars show the total number of core tensors in `Phase I` and `Phase II`, respectively; (b) Solid and hollow markers indicate the RSE achieved by the proposed algorithm after `Phase II` and by the vanilla TN-SS with $|V|$ matched to that obtained by the proposed algorithm. The red dotted line marks the $RSE = 10^{-6}$ threshold; (c) Light and dark bars denote the total number of evaluations in `Phase I` and `Phase II`, respectively; (d)-(f) Search dynamics of **Rand.** using TNGA, where blue and red regions correspond to `Phase I` and `Phase II`.

## 4.2 PARAMETER EFFICIENT FINE-TUNING (PEFT) FOR LLMS

We next demonstrate that our approach improves parameter efficiency in the PEFT task for LLMs. For disclaimer, this experiment is *not* intended to propose a new practical PEFT method, but rather to highlight how joint TN-SS itself contributes to parameter-efficient representations.

**Setup.** We build upon QuanTA (Chen et al., 2024a), a state-of-the-art PEFT method for LLMs utilizing TNO. In this experiment, we enhance it by applying our algorithms to optimize TNO structures, for the goal of reducing the number of fine-tuned parameters. We conduct the experiment on five commonsense reasoning datasets with Llama2-7B model (Touvron et al., 2023) (involving the TNO system of dimension $M = 32$), and follow most of the settings as the work (Chen et al., 2024a), where each fine-tuned weight with size $4096 \times 4096$ is recognized as a tensor of order-8 with size[3] $16 \times 8 \times 8 \times 4 \times 16 \times 8 \times 8 \times 4$. In our algorithm, we directly use the structure in QuanTA as the common graph $G$, from which we deploy the proposed algorithm from `Phase II`.

**Results.** Table 1 summarizes the test results, including baseline comparisons with LoRA (Hu et al., 2022), DoRA (Liu et al., 2024a), MixLoRA/MixDoRA (Li et al., 2024a), VeRA (Kopiczko et al., 2023) and VB-LoRA (Li et al., 2024b). Our approach achieves superior performance with the same or fewer fine-tuning parameters. Figure 3 (a) shows the search dynamics across five datasets, where joint TN-SS consistently reduces the parameter count throughout the search. Figure 3 (b) further illustrates the layer-wise parameter allocation on ARC-c, demonstrating that our approach allows *non-uniform* parameter distribution across layers, maximizing overall parameter efficiency. This observation is consistent with findings discussed in Section 1, which suggest that different weights within a system often exhibit varying structural complexities. More results are shown in Appendix C.4.

## 4.3 MEMORY-EFFICIENT QUANTUM CIRCUIT OPTIMIZATION

Quantum circuit optimization seeks to represent high-dimensional unitary operators using compact circuits of lower-dimensional unitary tensors. Mathematically, the task is equivalent to optimizing the structure for TNOs. More compact circuits naturally imply lower latency, less noise accumulation,

---

[3]Note that QuanTA slightly generalizes the definition of TNOs by varying the parameter $I$. We adopt the same experimental settings, as the variation in $I$ does not impact the search process of our algorithm.

Table 1: Test performance of PEFT. The results marked with "*" are from Li et al. (2024a). The details for QuanTA-6/4/2 are described in Appendix C.2.

| Method | #Params | PIQA | SIQA | OBQA | ARC-e | ARC-c | Avg. |
|---|---|---|---|---|---|---|---|
| LoRA* | 3.200% | 82.1 | 69.9 | 80.4 | 73.8 | 50.9 | 71.4 |
| DoRA* | 3.200% | 82.7 | 74.1 | 80.6 | 76.5 | 59.8 | 74.7 |
| MixLoRA* | 3.200% | 83.2 | 78.0 | 81.6 | 77.7 | 58.1 | 75.7 |
| MixDoRA* | 3.200% | 82.2 | 80.4 | 80.9 | 77.5 | 58.2 | 75.8 |
| VB-LoRA | 0.042% | 74.8 | 74.9 | 78.0 | 81.7 | 60.8 | 74.0 |
| QuanTA-6 | 0.041% | 79.9 | 75.9 | 80.0 | 84.8 | 63.3 | 76.8 |
| Ours | 0.031% | 80.4 | 76.2 | 80.2 | 84.7 | 63.7 | **77.0** |
| QuanTA-4 | 0.024% | 79.2 | 73.5 | 77.2 | 84.4 | 62.6 | 75.4 |
| Ours | 0.024% | 80.3 | 75.1 | 78.4 | 84.2 | 63.4 | **76.3** |
| VeRA | 0.018% | 77.5 | 68.3 | 72.4 | 79.3 | 53.7 | 70.2 |
| QuanTA-2 | 0.017% | 78.1 | 72.5 | 74.8 | 82.1 | 57.1 | 72.9 |
| Ours | 0.017% | 79.7 | 72.7 | 78.0 | 83.1 | 59.9 | **74.7** |

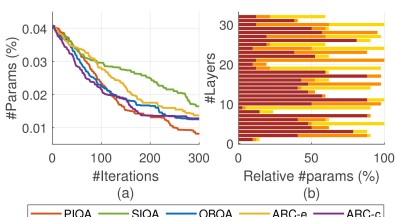

Figure 3: Param. reduction dynamics. (a) Number of param. over iterations; (b) Layer-wise parameter allocation on ARC-c, where the three colors indicate different stages in search.

Table 2: Result for quantum circuit optimization using joint TN-SS.

| Case | #Params of original opreator | Brick-wall (baseline) | | Classic QFT circuit | | Ours | |
|---|---|---|---|---|---|---|---|
| | | #Params | Fidelity↑ | #Params | Fidelity↑ | #Params | Fidelity↑ |
| Synthetic ($Q = 8$) | 65536 | 560 | $> 1 - 10^{-4}$ | / | / | 208 | $> 1 - 10^{-4}$ |
| Synthetic ($Q = 12$) | $1.68 \times 10^7$ | 528 | $> 1 - 10^{-4}$ | / | / | 176 | $> 1 - 10^{-4}$ |
| QFT ($Q = 4$) | 256 | 240 | $> 1 - 10^{-4}$ | 96 | $> 1 - 10^{-4}$ | 96 | $> 1 - 10^{-4}$ |
| QFT ($Q = 6$) | 4096 | 800 | 0.9859 | 240 | $> 1 - 10^{-4}$ | 288 | $> 1 - 10^{-4}$ |

and a more stable implementation in practice. Conventional designs, such as multi-layer brick-wall circuits, are often redundant and inefficient. Our proof-of-concept study (in modest scale for demonstration) shows that joint TN-SS can discover novel circuits with fewer core tensors, suggesting new paths for state preparation and quantum architecture search.

**Setup.** We evaluate both synthetic unitary operators (generated with Haar-random cores, $K = 2$, $I = 2$, and $Q = 8, 12$) and the quantum Fourier transform (QFT) (Camps et al., 2021; Chen et al., 2023), where the same settings of $K, I, Q$ are applied for QFT. For baselines, we use brick-wall TNOs and, for QFT, the classic hand-crafted QFT circuit. In our method, repeated brick-wall layers are treated as the common graph $G$, and structural diversity emerges through symmetry breaking.

**Results.** Table 2 shows that our approach consistently produces more compact TNO representations than the brick-wall baseline while maintaining fidelity above $1 - 10^{-4}$. For synthetic operators, we achieve up to a $3\times$ reduction in parameters. For QFT, our method recovers circuits with parameter counts on par with the classic design, yet *without SWAP gates*, a critical advantage for noise mitigation on quantum hardware. Figure 4 further illustrates this: for $Q = 4$, the result matches the conventional circuit (Camps et al., 2021; Chen et al., 2023), while for $Q = 6$ it discovers a distinctly different and more compact architecture using only local operators. These findings demonstrate that joint TN-SS not only advances efficiency in ML tasks but also provides a new path to discovering efficient quantum circuits. Full experimental details are given in Appendix D.

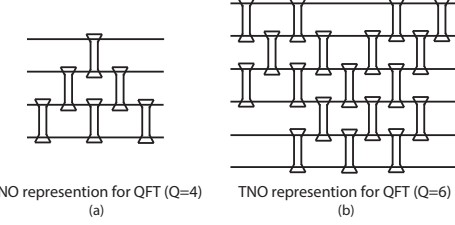

TNO representation for QFT (Q=4)
(a)

TNO representation for QFT (Q=6)
(b)

Figure 4: Optimized TNO representation for the QFT operators.

## 5 CONCLUDING REMARKS

In this work, we introduced *joint TN-SS*, a previously unexplored extension of tensor network structure search. Inspired by the principle of symmetry breaking, we proposed a simple yet effective algorithm to tackle the combinatorial complexity inherent in joint TN-SS. Extensive experiments across diverse tasks, including tensor decomposition, LLM fine-tuning, and quantum circuit optimization, demonstrate the effectiveness and efficiency of our approach. Joint TN-SS offers a systematic framework for model selection in tensorized ML and enables the automatic discovery of compact quantum circuits that reduce depth, noise, and latency in the NISQ regime. Conceptually, the symmetry-breaking viewpoint highlights a useful coarse-to-fine search principle: *starting from simpler, lower-dimensional structures can guide efficient solutions to harder problems.*

**Limitations.** Like most TN-SS methods, joint TN-SS is also a *meta*-algorithm, meaning that its overall scalability depends on a task-dependent inner optimization procedure. Our quantum-circuit experiments are currently limited to modest scales due to the memory cost of the inner optimization (see Appendix D.5 for hardware details). Extending joint TN-SS to larger qubit systems and more local operators settings is an important direction for future work. Additionally, we observed instability in widely used inner optimization methods, such as gradient-based and SVD-based approaches (Schollwöck, 2005), when applied to tensor decomposition and quantum circuit optimization. Improving the robustness of these optimization routines, and integrating them more tightly with joint TN-SS, remains a valuable avenue for further research.

## REPRODUCIBILITY STATEMENT

We have made substantial efforts to ensure the reproducibility of our work. Detailed descriptions of our proposed algorithm are provided in Section 3.2. Detailed settings for each experiment—including datasets, hyperparameter configurations, and implementation details—are presented in Appendix B, C and D. We have included the source code for tensor decomposition and LLM fine-tuning in the supplementary materials (the code for quantum circuit optimization has been provided in the rebuttal phase). For theoretical results (i.e., Proposition 3.1), the complete proof is presented in Appendix E. We encourage readers to refer to these sections for comprehensive information necessary to reproduce our work.

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

APPENDIX

## TABLE OF CONTENTS

## A  A PRACTICAL AND INTUITIVE INTRODUCTION TO JOINT TN-SS

Tensor network operators (TNOs) provide a structured way to represent large linear operators using many small tensors connected through contraction rules. Section 2.1 provides a mathematically precise formulation, while this appendix offers an intuitive, ML-friendly perspective using Einstein summation to illustrate how TNOs work in practice.

**From linear Operators to TNOs.**  In this paper, we focus on linear operators of the form $W \in \mathbb{F}^{I^Q \times I^Q}$, where $\mathbb{F}$ is either the real or complex field. Such operators commonly arise in ML. For example, in a Transformer layer, a weight matrix $W \in \mathbb{R}^{1024 \times 1024}$ for computing queries, keys, or values satisfies $I^Q = 1024$, with $I = 2$ and $Q = 10$ (or any factorization producing the correct dimension). If the dimensions of $W$ are not factorizable, a zero-padding trick is typically used.

Note that the cost of storing and computing $W$ commonly scales as $\mathcal{O}(I^{2Q})$ or even higher, which becomes expensive even for moderate $Q$. This cost dominates memory and computation in most of large-scale learning systems.

Fortunately, once we reshape $W$ into a tensor of dimension $\underbrace{I \times I \times \cdots \times I}_{\text{repeated } Q \text{ times}} \times \underbrace{I \times I \times \cdots \times I}_{\text{repeated } Q \text{ times}}$,

TNOs *reduce* the cost by factorizing $W$ into many small building blocks $\{V_i\}_{i=1}^{L}$ called *core tensors* in the main text. Following the circuit graph (focused in this paper), each core behaves like a small operator of shape $V_i \in I^K \times I^K$ where $K \ll Q$ in general. These cores are then composed to approximate or exactly represent the large global operator $W$.

**Implementation contraction with Einstein summation (Einsum).**  To represent the high-dimensional operator $W$, the building block $\{V_i\}_{i=1}^{L}$ should be multipled together. Unlike the normal matrix multiplication, the "multiplication" among tensors is called *contraction*, which is an extension of matrix multiplication, and can be easily implemented using Einsum notation (and the corresponding Python function) in practice.

Recall the corner illustration in Section 2.1 (reshown on the right-hand side). Subfigure (b) depicts four core tensors connected by shared edges in the circuit graph. We highlight that each edge corresponds to an index, and whenever two core tensors share an edge, they also share an index to be contracted. The contraction of the four-core TNO in subfigure (b) can be thus implemented as follows:

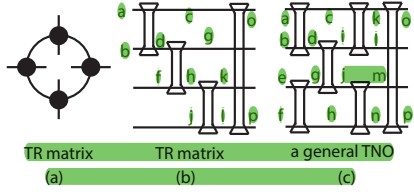

```
W = torch.einsum("abdc,dfgh,hjkl,clop -> abfjogkp", V_1, V_2, V_3, V_4)
```

Here, the torch tensors `V_1` to `V_4` correspond to the four cores from left to right shown in the graph. NOte that each core tensor has four indices, matching the number of edges attached to it in the graph, and the shared indices encode exactly which edges are contracted.

The TNO shown in subfigure (c), which contains six interconnected core tensors, can be also implemented in the same way:

```
W = torch.einsum(
    "abcd,efgh,dgij,cikl,jhmn,knop -> abefolmp",
    V_1, V_2, V_3, V_4, V_5, V_6
)
```

These examples illustrate the key idea: *each line in the circuit graph corresponds to a shared index in Einstein summation, and each core tensor becomes one argument in an* `einsum` *call.* The final operator $W$ is simply the result of contracting all shared indices.

**From TNOs to TNO systems.**  A TNO system, as defined in the main text, can be viewed intuitively as a collection of learnable weight operators $W$s, all sharing the same $I$ and $Q$ (i.e., the same input/output dimensionality), but differing in their internal tensor-network structure. Each operator may use a different number of core tensors and a different pattern of contractions, resulting in distinct

network topologies. In this way, a TNO system provides a unified framework for representing heterogeneous yet structurally related operators within a learning model. Consequently, joint TN-SS can be formulated naturally over this object by searching across the space of such interconnected TNO structures.

**TNs v.s. TNOs.** As discussed in Section equation 2.1, TNOs are a class of tensor networks whose connectivity can be expressed in a "circuit-like" topology. They generalize a wide range of widely used TN models by appropriately reconnecting low-order core tensors, including MP-S/MPO (Anselme Martin et al., 2024), isometric PEPS (Slattery & Clark, 2021), Tree Tensor Networks (Hierarchical Tucker) (Seitz et al., 2023), and MERA Berezutskii et al. (2025), and, with sufficiently many cores, they are known to universally approximate arbitrary linear operators (Barenco et al., 1995; Möttönen et al., 2004). However, TNOs do *not* cover all tensor networks; they form a structured subset, whereas general TN-SS methods may search arbitrary TN graphs but typically restrict the number of cores to match the tensor modes. In contrast, TNOs allow an arbitrary number of cores but impose a circuit-style constraint. We focus on TNOs for three reasons: they offer strong practical benefits in recent ML studies (Chen et al., 2024a; Li et al., 2025), they align naturally with quantum circuit representations and quantum ML, and structure search over TNOs remains technically unexplored and challenging due to these architectural constraints. We believe this restriction helps TN-SS discover novel yet *practically useful* TN models.

**Intuition of symmetry breaking.** The concept of symmetry breaking introduced in this paper highlights a simple, efficient, yet often overlooked principle: *the optimal solution to a difficult problem often lies close to a suboptimal solution found in a coarser, lower-dimensional search space.* In the context of joint TN-SS, this means that instead of directly searching over the vast combinatorial space of all possible tensor-network topologies, we first identify a good solution within a more symmetric structural family. Once such a coarse solution is obtained, we gradually relax the symmetry, allowing each operator to deviate from the shared structure through small, targeted perturbations.

This symmetry-breaking principle enables the search to escape overly restrictive structural templates while avoiding the computational explosion of full combinatorial search. In practice, it allows different TNOs in a system to evolve toward heterogeneous yet coordinated topologies that better reflect their distinct roles in the model, achieving both tractability and expressiveness.

**Physical connection to joint TN-SS.** The physics metaphor of symmetry breaking (SB) (shown in Fig. 1) directly informs our algorithmic design. In joint TN-SS, the challenge is the combinatorial explosion of possible structures as the number of TNOs $M$ grows. SB suggests a natural way to manage this complexity. Regarding the Higgs potential described in Eq. 4, the joint TN-SS consists of two regimes.

- *Symmetry regime* ($\mu > 0$) — all TNOs share a common structure, collapsing the search to standard TN-SS and giving a tractable starting point;
- *SB regime* ($\mu < 0$) — diversity is gradually introduced as the system "selects" different operator-specific structures, analogous to particles settling in different minima of the Mexican hat.

This phase transition, from enforced symmetry to controlled asymmetry, provides a principled mechanism to reduce the search space initially, then progressively enrich it. Instead of facing the full exponential complexity of unconstrained joint search, our algorithm leverages SB to unlock diversity only when needed, enabling efficient exploration of heterogeneous TNO structures.

# B    TECHNICAL APPENDICES WITH ADDITIONAL RESULTS FOR SYNTHETIC DATA

## B.1    DETAILS OF SYNTHETIC DATA GENERATION

Fig. 5 illustrates the four common TNO graphs $G$ with $Q = 5$. Table 3 and 4 summarize the structures (in VI matrix form) for generating the TNO systems used in our experiment. Specifically, each TNO system contains 4 TNOs, with each TNO generated by masking core tensors with locations 1 to 4

described in Table 3 and 4 from the given common graph. For example, the 2nd TNO for the stairs structure is generated by masking the 1st, 2nd, and 8th core tensors, which corresponds to removing the 1st, 2nd, and 8th columns from the VI matrix of the original stairs structure.

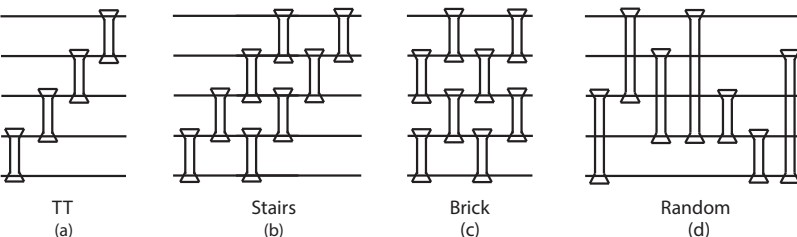

|  | TT (a) | Stairs (b) | Brick (c) | Random (d) |

Figure 5: Illustration of common graphs used in our experiment.

Once the TNO system is constructed, we sample each element in core tensors from i.i.d Gaussian distribution with zero mean and unit variance, and the (observed) synthetic tensor data is obtained by contracting the core tensors together.

Table 3: TNO generation for **TT**, **Stairs**, and **Brick** structures with $Q = 5, K = 2$.

| Structure | Common graph (VI matrix) | Mask locations 1 | Mask locations 2 | Mask locations 3 | Mask locations 4 |
|---|---|---|---|---|---|
| TT | $\begin{pmatrix} 4 & 3 & 2 & 1 \\ 5 & 4 & 3 & 2 \end{pmatrix}$ | 1 | 2 | 3 | 4 |
| Stairs | $\begin{pmatrix} 4 & 3 & 2 & 1 & 4 & 3 & 2 & 1 \\ 5 & 4 & 3 & 2 & 5 & 4 & 3 & 2 \end{pmatrix}$ | 1, 2, 3 | 1, 2, 8 | 3, 4, 6 | 1, 4, 7 |
| Brick | $\begin{pmatrix} 4 & 2 & 3 & 1 & 4 & 2 & 3 & 1 \\ 5 & 3 & 4 & 2 & 5 & 3 & 4 & 2 \end{pmatrix}$ | 1, 2, 8 | 1, 6, 8 | 2, 4, 7 | 6, 7, 8 |

Table 4: TNO generation for **Rand.** structures with different $Q$ ($K = 2$).

| Q | Common graph (VI matrix) | Mask locations 1 | Mask locations 2 | Mask locations 3 | Mask locations 4 |
|---|---|---|---|---|---|
| 4 | $\begin{pmatrix} 3 & 1 & 1 & 2 & 2 \\ 4 & 2 & 3 & 4 & 3 \end{pmatrix}$ | 1 | 2 | 3 | 4 |
| 5 | $\begin{pmatrix} 3 & 1 & 2 & 1 & 3 & 4 & 2 \\ 5 & 3 & 4 & 4 & 4 & 5 & 5 \end{pmatrix}$ | 1, 3 | 2, 3 | 1, 4 | 2, 4 |
| 6 | $\begin{pmatrix} 3 & 1 & 1 & 3 & 2 & 4 & 3 & 1 \\ 5 & 6 & 4 & 6 & 4 & 5 & 4 & 3 \end{pmatrix}$ | 1, 2, 3 | 2, 3, 6 | 1, 4, 5 | 4, 6, 8 |
| 8 | $\begin{pmatrix} 6 & 5 & 2 & 1 & 5 & 1 & 4 & 3 & 1 & 3 \\ 7 & 6 & 5 & 2 & 8 & 4 & 7 & 4 & 3 & 6 \end{pmatrix}$ | 1, 2, 3 | 2, 3, 6 | 1, 4, 5 | 4, 6, 8 |

### B.2 TN-SS ALGORITHMS FOR PHASE I

The five joint TN-SS algorithms with default parameter settings used in Phase I are summarized in Algo. 2-5, which can be seen as extensions of the TNGA (Li & Sun, 2020), TNLS (Li et al., 2022), TnALE (Li et al., 2023) and Greedy (Hashemizadeh et al., 2020) methods, respectively. To align with the TNO models targeted in this work, we slightly modified these algorithms but tried our best to keep the searching dynamics unchanged.

In TNO, the adjacent core tensors can be merged if they have the same connectivity (i.e., the adjacent columns in Eq. (3) are the same). For the proposed TN-SS algorithms, once a new TNO graph is generated, we always check if the adjacent core tensors can be merged. If they are mergeable, one of the duplicated core tensors will be replaced with a new one with different connectivity. This procedure is repeated until all core tensors in the new TNO graph can not be merged.

For TNLS and TnALE, the neighborhood of a TNO $G$ is denoted as $\mathcal{N}(G, s)$, and is defined as the set of graphs that has the same connectivity of all core tensors except for the $s$-th one. In the view of

---

**Algorithm 2** Genetic Algorithm for joint TN-SS (TNGA) in `Phase I`

---

1: **Input:**
2: $\mathcal{X}^M$         ▷ Group of input tensors
3: $Q, K, I$         ▷ parameters for TNO
4: $L$         ▷ order for the common graph
5: $S = 100$         ▷ population size of individuals in each generation
6: $C_{max} = 20$         ▷ maximum number of generations
7: $f(\cdot) = \min_{\mathcal{X}^M \in tno(\cdot)} \pi_D(\mathcal{X}^M)$         ▷ loss function
8: $\eta_1 = 5 \times 10^{-10}$         ▷ tolerance
9: $\varepsilon = 0.2$         ▷ elimination parameter
10: $p(r) = \max\left\{0.01, \ln(200/(10^{-2} + 5r))\right\}$
11: **Algorithm:**
12: randomly generate $\{G_s\}_{s=1}^S \in \{G(V, E) : G \in \mathbb{G}, |V| = L\}$         ▷ parent Initialization
13: **for** $t = 1$ **to** $C_{max}$ **do**
14:      $f^s = f(G_s)$ for all $s \in [S]$         ▷ fitness evaluation
15:      $\hat{G} = \arg\min_{G_s : s \in [S]} l^s$
16:      $\hat{f} = f(\hat{G})$
17:      **if** $f(\hat{G}) < \eta_1$ **then**
18:          *break*
19:      **end if**
20:      $\{r_s\}_{s=1}^S = \text{rank}(\{f^s\}_{s=1}^S)$         ▷ get rank of the individuals by fitness evaluation
21:      $\mathbb{G}_p = \{(G_s, p(r_s)) : s \in [S], r_s \leq \lceil(1 - \varepsilon)S\rceil\}$    ▷ eliminate the $\varepsilon \times 100\%$ individuals with worst fitness and compute the sampling probability $p$ on remaining individuals
22:      **for** $s = 1$ **to** $S$ **do**
23:          $G_{p1}, G_{p2} \leftarrow \text{sample}(\mathbb{G}_p)$     ▷ select parents with probabilities $p$ and with replacement
24:          $G_s \leftarrow \text{crossover\&mutate}(G_{p1}, G_{p2})$         ▷ generate child from the parents
25:      **end for**
26:      $\{G_s\}_{s=1}^S \leftarrow \text{deduplicate\&fill}(\{G_s\}_{s=1}^S)$   ▷ remove individuals with duplicated graphs and generate new individuals to replace them
27: **end for**
28: **Output:** $\hat{G}, \hat{f}$

---

---

**Algorithm 3** Local Sampling algorithm for joint TN-SS (TNLS) in `Phase I`

---

1: **Input:**
2: $\mathcal{X}^M, Q, K, I, f(\cdot)$ ▷ Group of input tensors and system parameters
3: $L, C_{max} = 40, \theta = 0.5, \eta_1 = 5 \times 10^{-10}$ ▷ algorithm parameters. $\theta$: sampling ratio
4: **Algorithm:**
5: initial $\hat{G} \in \{G(V,E) : G \in \mathbb{G}, |V| = L\}$ ▷ Initialize the TNO
6: $\hat{f} = f(\hat{G})$ ▷ Evaluation on initial TNO
7: **for** $t = 1$ **to** $C_{max}$ **do**
8:     $\mathbb{G}_s = \varnothing$ ▷ Initialize the sampling set
9:     **for** $s = 1$ **to** $L$ **do**
10:         $\mathbb{G}_s \leftarrow \mathbb{G}_s \cup \{G : G(V,E) \in \mathbb{G}, |V| = L, G \in \mathcal{N}(\hat{G}, s)\}$ ▷ Add neighborhood TNOs to the sampling set
11:     **end for**
12:     $\mathbb{G}_s \leftarrow \text{random\_sampling}(\mathbb{G}_s, \theta)$ ▷ Randomly sample $\theta \times 100\%$ TNOs from the sampling set
13:     **if** $\min_{G \in \mathbb{G}_s} f(G) < \hat{f}$ **then** ▷ Evaluation on sampled TNOs and update the best TNO if the condition satisfies
14:         $\hat{G} = \arg\min_{G \in \mathbb{G}_s} f(G)$
15:         $\hat{f} = f(\hat{G})$ ▷ Update the estimated TNO graph and corresponding loss
16:     **end if**
17:     **if** $\hat{f} < \eta_1$ **then**
18:         *break*
19:     **end if**
20: **end for**
21: **Output:** $\hat{G}, \hat{f}$

---

**Algorithm 4** Alternating Local Enumeration algorithm for joint TN-SS (TnALE) in `Phase I`

---

1: **Input:**
2: $\mathcal{X}^M, Q, K, I, f(\cdot)$ ▷ Group of input tensors and system parameters
3: $L, D = 2, \eta_1 = 5 \times 10^{-10}$ ▷ algorithm parameters. $D$: round-trips of ALE
4: **Algorithm:**
5: initial $\hat{G} \in \{G(V,E) : G \in \mathbb{G}, |V| = L\}$ ▷ Initialize the TNO
6: $\hat{f} = f(\hat{G})$ ▷ Evaluation on initial TNO
7: **for** $t = 1$ **to** $D$ **do**
8:     **for** $s = 1$ **to** $L$ **do** ▷ Forward trip
9:         $\mathbb{G}_s = \{G : G(V,E) \in \mathbb{G}, |V| = L, G \in \mathcal{N}(\hat{G}, s)\}$ ▷ Add neighborhood TNOs to the sampling set
10:         **if** $\min_{G \in \mathbb{G}_s} f(G) < \hat{f}$ **then** ▷ Evaluation on sampled TNOs and update the best TNO if the condition satisfies
11:             $\hat{G} = \arg\min_{G \in \mathbb{G}_s} f(G)$
12:             $\hat{f} = f(\hat{G})$
13:         **end if**
14:         **if** $\hat{f} < \eta_1$ **then**
15:             *break*
16:         **end if**
17:     **end for**
18:     **for** $s = L - 1$ **to** $2$ **do** ▷ Backward trip
19:         Repeat steps 9-15
20:     **end for**
21: **end for**
22: **Output:** $\hat{G}, \hat{f}$

---

---

**Algorithm 5** Greedy algorithm for joint TN-SS in `Phase I`

---

1: **Input:**
2: $\mathcal{X}^M, Q, K, I, f(\cdot)$          ▷ Group of input tensors and system parameters
3: $L, \eta_1 = 5 \times 10^{-10}$                  ▷ algorithm parameters
4: **Algorithm:**
5: initial $\hat{G} = \varnothing$                   ▷ Initialize an empty TNO
6: **for** $s = 1$ **to** $L$ **do**
7:      $\mathbb{G}_s = \{G : G(V, E) \in \mathbb{G}, |V| = s, G \in \mathcal{N}^+(\hat{G})\}$    ▷ Add neighborhood TNOs to the sampling set
8:      $\hat{G} = \arg\min_{G \in \mathbb{G}_s} f(G)$         ▷ Find the best TNO with minimum loss
9:      $\hat{f} = f(\hat{G})$                    ▷ Update the loss
10:      **if** $\hat{f} < \eta_1$ **then**
11:          *break*
12:      **end if**
13: **end for**
14: **Output:** $\hat{G}, \hat{f}$

---

the VI matrix, $\mathcal{N}(G, s)$ is generated by replacing the $s$-th column of the VI matrix of $G$ with columns that represent all other possible connectivity.

Further, in the Greedy algorithm, the TNO is gradually generated with the graph order from 0 to $L$. At each iteration, starting from the previous TNO graph $G$, we add a core tensor to the right of the previous TNO, which is equivalent to adding a new column to the right of the VI matrix of $G$. The TNO set with all possible connectivity is denoted as $\mathcal{N}^+(G)$ in Algo. 5, and the best TNO for the next iteration is chosen as the one with the lowest RSE.

### B.3 Tensor decomposition for TNO

Given the TNO graph, we apply Adam optimizer (Kingma & Ba, 2014) to perform tensor decomposition, which is commonly used in existing TN-SS methods. Specifically, given a TNO graph and the observed tensor $\mathcal{X}$, we initialize each core tensor from Gaussian distribution with zero mean and variance 0.3. The contraction expression of core tensors is constructed using `einsum` function similar to that in (Chen et al., 2024a). The relative squared error (RSE) is used as the loss function in Eq. (2). Specifically, given the generated synthetic tensors $D := \{\mathcal{D}_1, \mathcal{D}_2, \ldots, \mathcal{D}_M\}$ and the approximated tensors $\mathcal{X}^M := \{\mathcal{X}_1, \mathcal{X}_2, \ldots, \mathcal{X}_M\}$, the RSE is defined as $\pi_D(\mathcal{X}^M) = \sum_{m=1}^M \|\mathcal{X}_m - \mathcal{D}_m\|_F^2 / \sum_{m=1}^M \|\mathcal{D}_m\|_F^2$, where $\|\cdot\|_F$ is the Frobenius norm. The learning rate of Adam is set to $10^{-2}$ and the maximum iteration number is set to 1500.

For each evaluation, the RSE is measured over 10 runs with different initializations of core tensors. Specifically, we select the minimum squared approximation error for each tensor over 10 runs and compute the overall RSE of four tensors. In `Phase I`, the tolerance $\eta_1$ for Algo. 2-5 is set to $5 \times 10^{-10}$ to determine if the common graph $G$ is searched, and the initialization strategies are the same as existing TN-SS methods. The setting of common graph order $L$ is determined differently for each data, following a basic principle that $L$ should be relatively large to achieve a satisfactory approximation error in `Phase I`. While in `Phase II`, the RSE is compared with tolerance $\eta_2 = 10^{-6}$ to determine if the core tensor should be masked.

### B.4 Implementation

In our experiments on synthetic data, we run all algorithms on a cluster of NVIDIA V100 GPUs alongside an Intel Xeon E5-2690 CPU node. Specifically, the CPU node handles data input, applies the TNO generation procedures, and distributes the sampled TNOs across the GPUs. Each GPU then performs the tensor decomposition for a given TNO and returns its loss. After each iteration, the CPU node collects these loss values and generates new TNOs according to the specific algorithm for the subsequent procedure.

Table 5: Results of the TD experiment. The RSE, total number of core tensors (shown in round brackets), and the corresponding number of evaluations (shown in square brackets) are presented for `Phase I` (top block) and `Phase II` (middle block). The bottom block shows the RSE of TN-SS methods for the same graph orders as `Phase II`.

| Method | TT | Stairs | Brick | Rand. |
|---|---|---|---|---|
| GREEDY | $< 5 \times 10^{-10}$ (24) [55] | $2.5 \times 10^{-7}$ (40) [91] | $7.8 \times 10^{-6}$ (40) [91] | 0.0426 (40) [91] |
| TnALE | $< 5 \times 10^{-10}$ (24) [109] | $< 5 \times 10^{-10}$ (40) [250] | $< 5 \times 10^{-10}$ (40) [94] | $2.12 \times 10^{-5}$ (40) [265] |
| TNLS | $< 5 \times 10^{-10}$ (24) [89] | $< 5 \times 10^{-10}$ (40) [217] | $< 5 \times 10^{-10}$ (40) [613] | $2.25 \times 10^{-6}$ (40) [1441] |
| TNGA | $< 5 \times 10^{-10}$ (24) [300] | $4 \times 10^{-8}$ (40) [2000] | $< 5 \times 10^{-10}$ (40) [800] | $< 5 \times 10^{-10}$ (40) [900] |
| GREEDY-SB | $< 5 \times 10^{-10}$ (12) [49] | $< 5 \times 10^{-10}$ (20) [140] | $< 5 \times 10^{-10}$ (24) [142] | 0.0426 (40) [40] |
| TnALE-SB | $< 5 \times 10^{-10}$ (12) [61] | $< 5 \times 10^{-10}$ (32) [82] | $1 \times 10^{-8}$ (32) [103] | $2.12 \times 10^{-5}$ (40) [40] |
| TNLS-SB | $< 5 \times 10^{-10}$ (20) [50] | $< 5 \times 10^{-10}$ (29) [136] | $< 5 \times 10^{-10}$ (22) [136] | $< 5 \times 10^{-10}$ (38) [66] |
| TNGA-SB | $< 5 \times 10^{-10}$ (12) [48] | $< 5 \times 10^{-10}$ (22) [147] | $1 \times 10^{-8}$ (26) [110] | $< 5 \times 10^{-10}$ (20) [112] |
| *RSE of only optimizing G with the $|V|$ aligned to the results in* `Phase II` | | | | |
| GREEDY | 0.3568 (12) [28] | 0.2032 (20) [46] | 0.1838 (24) [55] | 0.0426 (40) [91] |
| TnALE | 0.3637 (12) [66] | 0.0094 (32) [208] | 0.0133 (32) [208] | $2.12 \times 10^{-5}$ (40) [265] |
| TNLS | $< 5 \times 10^{-10}$ (20) [111] | 0.0045 (32) [1161] | 0.1118 (24) [881] | $2.25 \times 10^{-6}$ (40) [1441] |
| TNGA | 0.2907 (12) [2000] | 0.0145 (24) [2000] | 0.0243 (28) [2000] | 0.1050 (20) [2000] |

Table 6: Results on different $Q$ (random structure with $I = 2$).

| Method | $Q = 4$ | $Q = 5$ | $Q = 6$ | $Q = 7$ | $Q = 8$ |
|---|---|---|---|---|---|
| TNGA ($S$=100) | $< 5 \times 10^{-10}$ (40) [300] | $< 5 \times 10^{-10}$ (50) [600] | $< 7.4 \times 10^{-5}$ (50) [600] | 0.0048 (60) [1500] | 0.0312 (60) [600] |
| TNGA-SB | $< 5 \times 10^{-10}$ (16) [62] | $< 5 \times 10^{-10}$ (39) [40] | $< 7.4 \times 10^{-5}$ (50) [50] | 0.0048 (60) [60] | 0.0312 (60) [60] |
| TNGA ($S$=200) | – | – | $< 5 \times 10^{-10}$ (50) [1200] | $< 5 \times 10^{-10}$ (60) [4000] | 0.0237 (60) [3400] |
| TNGA-SB | – | – | $< 5 \times 10^{-10}$ (25) [178] | $< 5 \times 10^{-10}$ (32) [152] | 0237 (60) [60] |
| TNGA ($S$=300) | – | – | – | – | $< 5 \times 10^{-10}$ (60) [2100] |
| TNGA-SB | – | – | – | – | $< 5 \times 10^{-10}$ (34) [143] |

## B.5 ADDITIONAL EXPERIMENTAL RESULTS ON SYNTHETIC DATA

As a detailed version of Figure 2 (a)-(c), Table 5 reports all the RSE, the total number of core tensors, and the total number of evaluations for synthetic data. The algorithms with suffix '-SB' denote that Algo.1 is applied for `Phase II`. We should remark that, although Greedy and TNLS fail to meet the $RSE \leq 10^{-6}$ for **Stairs** and **Rand.** in `Phase II`, respectively, their RSE values are close to $10^{-6}$ such that the `Phase II` can still work to reach the RSE lower than $10^{-6}$.

We conduct additional experiments to verify the performance on different TNO parameters $Q$ and $I$. The **Rand.** structures with $Q = [4, 5, 6, 7, 8]$ (see Table.4) are applied as common graphs for data generation, and the TNGA method is applied for `Phase I`. Table. 6 shows the performance under different $Q$. We gradually increase the population size parameter $S$ from 100 until the RSE reaches $\leq 10^{-6}$. The results verify the effectiveness of the proposed method for variant selections of $Q$. Further, the $S$ needs to be set to a larger value as $Q$ increases due to the increasing searching space of the common graph. Consequently, the total number of evaluations in both `Phase I` and `Phase II` grows with $Q$ increases. Further, the decomposition performance under TNO with different $I$ is depicted in Fig. 7. It can be seen that the proposed algorithm works well on TNO with varying selections of $I$.

We also investigate the performance of different common graph orders $L$ used in `Phase I`. Fig. 6 depicts the performance under different $L$ for the proposed method. The results imply that $L$ works well when $L$ is slightly larger than groundtruth $L^* = 7$. It can be also seen that if the $L$ is

Table 7: RSE, total number of core tensors (round brackets), and the corresponding number of evaluations (square brackets) on different $I$ (**Rand.** structure with $Q = 5$).

| Method | $I = 2$ | $I = 4$ | $I = 6$ |
|---|---|---|---|
| TNGA | $< 5 \times 10^{-10}$ (32) [100] | $< 5 \times 10^{-10}$ (32) [1000] | $< 5 \times 10^{-10}$ (32) [800] |
| TNGA-SB | $< 5 \times 10^{-10}$ (16) [70] | $< 5 \times 10^{-10}$ (31) [32] | $6.4 \times 10^{-7}$ (22) [69] |

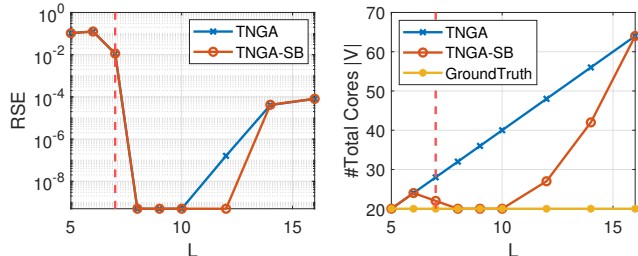

Figure 6: RSE (left) and the total number of core tensors (right) under **Rand.** structure ($Q = 5, K = 2, I = 2$) using different parameter $L$. Dotted red line: order of groundtruth **Rand.** structure.

Table 8: RSE and the corresponding number of evaluations (square brackets) on different $M$ (**Rand.** structure with $Q = 5$).

| Method | $M = 2$ | $M = 4$ | $M = 6$ | $M = 8$ |
|---|---|---|---|---|
| Vanilla TN-SS | $< 5 \times 10^{-10}$ [700] | $< 5 \times 10^{-10}$ [3800] | $2.9 \times 10^{-6}$ [6000] | $1.5 \times 10^{-4}$ |
| Ours | $< 5 \times 10^{-10}$ [574] | $< 5 \times 10^{-10}$ [823] | $< 5 \times 10^{-10}$ [2027] | $< 5 \times 10^{-10}$ [2226] |

underestimated (i.e. $L < 7$), the approximation always fails as they can not find the common graph with $L^* = 7$. When $L = 7$, the approximation still fails as the TNGA method may fail to find the common graph with a limited number of evaluations. On the other hand, if the $L$ is too large (i.e., $L \geq 14$ in this experiment), the approximation also fails due to the optimization problem. In practice, as the $L^*$ is always not known, we may conduct TNGA on various selections of $L$ and choose a proper one that achieves a satisfactory approximation error.

Further, the efficiency of the proposed joint TN-SS method under different $M$ is verified. TNGA is employed as the TN-SS algorithm for Phase I. The maximum number of evaluations $C_{max}$ is $1000 \times M$. Table 8 presents the results of tensor decomposition for various values of $M$. We add the vanilla TN-SS as the comparison, in which each TNO is optimized independently without searching a common graph. As shown, for $M = 4$, the vanilla TN-SS requires significantly more evaluations than the proposed method to search TNOs that can successfully approximate the tensors. As $M$ increases to 6 and 8, the vanilla TN-SS reaches the maximum evaluation $C_{max}$ but still fails to achieve the desired approximation accuracy of $RSE \leq 10^{-6}$. In contrast, although increasing slightly raises computational complexity for the proposed method, it substantially enhances search efficiency and final performance due to the presence of shared structural patterns across TNOs. This confirms that symmetry breaking enables more effective exploration of the TNO structure space—especially when the TNOs are related (e.g., originating from similar data or architectural settings).

Finally, the sensitivity of different selections of the tolerance parameter $\eta_2$ in Phase II is analyzed. The experiment is conducted on **Rand.** structure with $Q = 5$, and TNGA is applied for Phase I. Table 9 reports the performance in terms of different $\eta_2$. These results show that the method is robust in a wide range of $\eta_2$ as long as the task has an internal tensor network structure. As expected, larger values give a more compact structure, and smaller values improve accuracy with more structure preserved.

### B.6 Computational Analysis of the Proposed Algorithm

The complexity of the proposed joint TN-SS algorithm contains two parts, i.e., Phase I and Phase II. For Phase II, as illustrated in Algo. 1, each iteration requires $|V|$ evaluations in the worst case, where $|V|$ is the total number of core tensors in the TNO system. This value typically scales linearly with the number of TNO in the system (*i.e.*, $M$). Thus, even in the worst case, the number of evaluations in Phase II grows linearly with the size of the system.

The computational cost of the proposed joint TN-SS method mainly depends on the TN-SS algorithm used for Phase I. For example, when taking TnALE in Phase I into account, the overall cost for the whole approach is $\mathcal{O}(LM)$ per iteration. If a different TN-SS algorithm is used as Phase I,

Table 9: RSE, total number of core tensors (round brackets), and the corresponding number of evaluations (square brackets) on different $\eta_2$ (**Rand.** structure with $Q = 5$).

| | $\eta_2 = 1$ | $\eta_2 = 0.01$ | $\eta_2 = 10^{-4}$ | $\eta_2 = 10^{-6}$ | $\eta_2 = 10^{-8}$ |
|---|---|---|---|---|---|
| TNGA-SB | 0.9926 (4) [41] | $< 5 \times 10^{-10}$ (20) [83] | $< 5 \times 10^{-10}$ (20) [102] | $< 5 \times 10^{-10}$ (20) [112] | $< 5 \times 10^{-10}$ (20) [87] |

additional factors naturally arise, such as the population size in TNGA or the number of sampled structures in TNLS.

We should note that the aforementioned analysis focuses on the cost of evaluations, while the proposed joint TN-SS is essentially a "meta-algorithm". Its total complexity additionally depends on the inner minimization in (5), which varies by task (tensor decomposition, PEFT, quantum circuit optimization).

## C   TECHNICAL APPENDICES WITH ADDITIONAL RESULTS FOR LLMS FINE-TUNING

### C.1   COMMONSENSE REASONING DATASETS

Table 10 presents detailed information about the datasets used in our experiments. Specifically, (Hu et al., 2023) originally collected all datasets including the training set **Train** and the test set **Test**. In our experiments, we randomly select 3000 and 400 data from the original training set as the training data **Train split** and validation data **Valid split**, respectively. We should note that, for ARC-Easy and ARC-Challenge datasets, the training data for our experiments are 1825 and 720 due to insufficient original training data. The fine-tuning performance is evaluated on the test set **Test**.

Table 10: Description of common sense reasoning datasets used in experiments.

| Dataset name | Domain | # Train | # Train split | # Valid split | # Test |
|---|---|---|---|---|---|
| PIQA | Physical Interaction | 16113 | 3000 | 400 | 1838 |
| SIQA | Social Interaction | 33410 | 3000 | 400 | 1954 |
| OBQA | Science Facts | 4957 | 3000 | 400 | 500 |
| ARC-Easy (ARC-e) | Natural Science | 2251 | 1851 | 400 | 2376 |
| ARC-Challenge (ARC-c) | Natural Science | 1119 | 719 | 400 | 1172 |

### C.2   PEFT FOR TNO

For each dataset, the fine-tuning is performed on **Train split** described in Table 10. The QuanTA model (Chen et al., 2024a) is adapted to our experiments with modifications that can handle different TNOs for each transformer layer. The training parameters used in experiments are shown in Table 11. The baseline algorithms QuanTA-6/4/2 that use the same TNO for all layers are shown in Table 12. Specifically, QuanTA-6 corresponds to the original QuanTA algorithm proposed in (Chen et al., 2024a).

In our experiment, we directly use the TNO of QuanTA-6 as the common graph $G$ and deploy the proposed algorithm for `Phase II`. For each evaluation, given a TNO system consisting of the TNOs for all layers, a training procedure with 3 epochs is conducted on **Train split**, following the inference on both **Valid split** and **Test**. Instead of using training loss as the loss function for $\pi_D(\cdot)$ in Algo. 1, we directly set the loss $f(\cdot)$ as $1 - acc_v$ where $acc_v$ is the accuracy on **Valid split**. The tolerance $\eta_2$ for each dataset is set according to the validation accuracy on TNO using QuanTA-6. Specifically, the $\eta_2$ for PIQA, SIQA, OBQA, ARC-e and ARC-c are set to $0.20, 0.23, 0.21, 0.15, 0.38$, respectively.

### C.3   IMPLEMENTATION

In our experiments on PEFT for LLMs, training is carried out on two NVIDIA RTX A6000 GPUs (each with 48 GB of memory). Similar to synthetic data, an Intel Xeon E5-2690 CPU node manages results collection and TNO generation.

Table 11: Hyperparameter Settings for LLM Fine-tuning.

| Hyperparameters | QuanTA/Ours | VeRA | VB-LoRA |
|---|---|---|---|
| Num of Epochs | 3 | | |
| Batch Size | 4 | | |
| Optimizer | AdamW | | |
| Scheduler | Linear Scheduler | | |
| Weight Decay | 0 | | |
| Dropout | 0 | | |
| Learning Rate | 1e-4 | 1e-3 | 1e-3 |
| Rank | - | 1024 | 4 |
| Vector Length | - | - | 256 |
| Vector Bank Size | - | - | 90 |
| Modules | (q_proj v_proj) | default | default |

Table 12: Different TNO structures of QuanTA used in experiments.

| Method | TNO for all layers (in VI matrix form) |
|---|---|
| QuanTA-6 | $\begin{pmatrix} 3 & 2 & 1 & 2 & 1 & 1 \\ 4 & 4 & 4 & 3 & 3 & 2 \end{pmatrix}$ |
| QuanTA-4 | $\begin{pmatrix} 3 & 2 & 1 & 1 \\ 4 & 3 & 2 & 4 \end{pmatrix}$ |
| QuanTA-2 | $\begin{pmatrix} 3 & 1 \\ 4 & 2 \end{pmatrix}$ |

## C.4 ADDITIONAL EXPERIMENTAL RESULTS ON LLM FINE-TUNING

Fig. 7 depicts the dynamic information during the `Phase II` on five datasets. (a)-(b) report the average accuracy degradation ($\times 100\%$) on **Test** and **Valid split**, respectively. The curves with light color in the background denote the accuracy at each iteration, and the curves with dark color foreground show the average accuracy smoothed with window size 20. (c)-(d) show the total number of core tensors and parameters at each iteration. It should be noted that the accuracy degradation on **Valid split** is bounded due to the settings of tolerance parameter $\eta_2$. As can be seen, for all datasets except SIQA, the total number of core tensors and parameters reduce quickly at the first 150 iterations and then slow down due to the setting of the tolerance $\eta_2$.

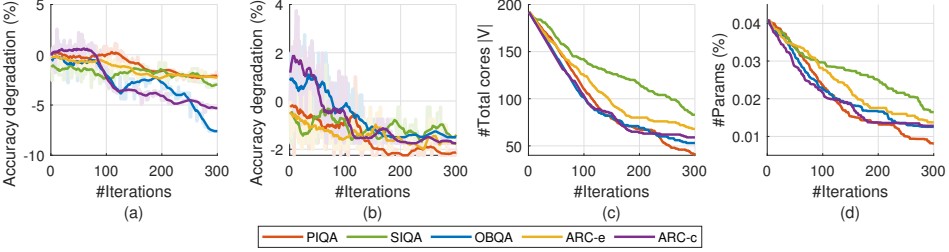

Figure 7: Accuracy degradation on test data and validation data over iterations (a,b). Total number of core tensors and #Params over iterations (c, d);

Fig. 8 shows the number of parameters on each transformer layer compared with QuanTA-6. The yellow, orange, and brown bars correspond to the three settings with the total number of parameters $0.031\%, 0.024\%$, and $0.017\%$, respectively. Further, the VI matrices of corresponding TNOs for 8/16/24/32 layers with the total number of parameters $0.024\%$ are shown in Table 13. The results imply the asymmetric property of TNO across different layers and different datasets.

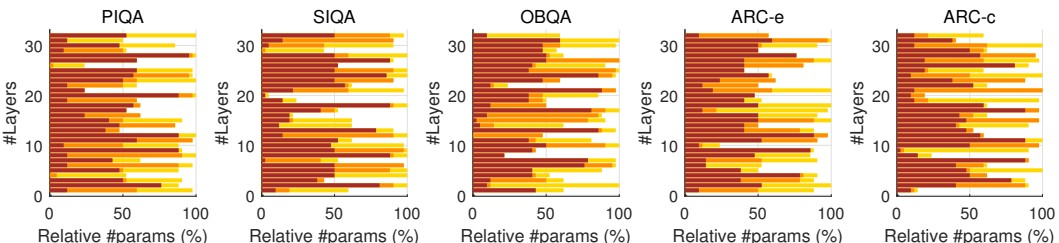

Figure 8: #Params percentage relative to QuanTA-6 per layer of Llama2-7B on five datasets. Yellow, orange, and brown bars indicate #Params of 0.031%, 0.024%, and 0.017%, respectively.

Table 13: TNO structures (in VI matrix form) of different transformer layers with #Params $0.024\%$.

| Layer | PIQA | SIQA | OBQA | ARC-e | ARC-c |
|---|---|---|---|---|---|
| 8 | $\begin{pmatrix} 3 & 2 & 1 & 1 & 1 \\ 4 & 4 & 4 & 3 & 2 \end{pmatrix}$ | $\begin{pmatrix} 3 & 2 & 1 & 1 & 1 \\ 4 & 4 & 4 & 3 & 2 \end{pmatrix}$ | $\begin{pmatrix} 3 & 1 & 2 \\ 4 & 4 & 3 \end{pmatrix}$ | $\begin{pmatrix} 1 & 1 \\ 4 & 2 \end{pmatrix}$ | $\begin{pmatrix} 3 & 2 & 2 \\ 4 & 4 & 3 \end{pmatrix}$ |
| 16 | $\begin{pmatrix} 3 & 2 & 1 & 2 & 1 \\ 4 & 4 & 4 & 3 & 3 \end{pmatrix}$ | $\begin{pmatrix} 1 & 2 \\ 4 & 3 \end{pmatrix}$ | $\begin{pmatrix} 1 & 1 & 1 \\ 4 & 3 & 2 \end{pmatrix}$ | $\begin{pmatrix} 2 & 1 & 1 \\ 4 & 4 & 3 \end{pmatrix}$ | $\begin{pmatrix} 3 & 2 & 1 & 1 \\ 4 & 4 & 4 & 3 \end{pmatrix}$ |
| 24 | $\begin{pmatrix} 1 & 2 & 1 & 1 \\ 4 & 3 & 3 & 2 \end{pmatrix}$ | $\begin{pmatrix} 2 & 1 & 1 \\ 3 & 3 & 2 \end{pmatrix}$ | $\begin{pmatrix} 1 & 2 & 1 & 1 \\ 4 & 3 & 3 & 2 \end{pmatrix}$ | $\begin{pmatrix} 1 & 2 & 1 \\ 4 & 3 & 2 \end{pmatrix}$ | $\begin{pmatrix} 3 & 2 & 1 \\ 4 & 3 & 2 \end{pmatrix}$ |
| 32 | $\begin{pmatrix} 3 & 2 & 1 & 1 \\ 4 & 4 & 4 & 3 \end{pmatrix}$ | $\begin{pmatrix} 3 & 2 & 1 & 1 \\ 4 & 3 & 3 & 2 \end{pmatrix}$ | $\begin{pmatrix} 2 \\ 3 \end{pmatrix}$ | $\begin{pmatrix} 1 & 2 & 1 \\ 4 & 3 & 3 \end{pmatrix}$ | $\begin{pmatrix} 2 & 1 & 2 \\ 4 & 4 & 3 \end{pmatrix}$ |

# D  TECHNICAL APPENDICES WITH ADDITIONAL RESULTS FOR REPRESENTATION OF QUANTUM CIRCUIT OPTIMIZATION

## D.1  PROBLEM SETTINGS AND SOLUTIONS

The goal of quantum circuit optimization is to approximate a quantum operator $U$ by a quantum circuit composed of several quantum gates. Following the common setting in quantum computing, we assume that each two-qubit gate operates on adjacent qubits in a 1D chain. As shown in Fig. 9, a 4 qubits quantum operator $U$ (a) can be represented as a quantum circuit with five two-qubit gates (b). In practice, finding a quantum circuit like (b) is challenging. An efficient way that avoid circuit structure searching is directly using the quantum circuits with a brick-wall structure with 3 blocks, as shown in Fig. 9 (c). However, the quantum circuits constructed using brick-wall structures are not always compact due to the redundancy of quantum gates (e.g., the gates in the red-dotted boxes in Fig. 9 (c)). In this work, we aim to efficiently search for a more compact representation with fewer quantum gates compared to the brick-wall structure.

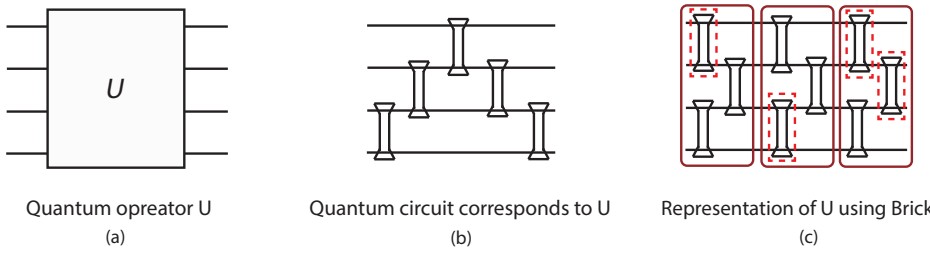

Quantum opreator U
(a)

Quantum circuit corresponds to U
(b)

Representation of U using Brick
(c)

Figure 9: Illustration of the representation of a quantum operator.

## D.2 GENERATION OF SYNTHETIC QUANTUM OPERATORS

In our experiment, we consider synthetic quantum operators with dimension $I = 2$ and qubit counts $Q = 4, 8, 12$. For each qubit count, a quantum circuit using a brick-wall structure with $M$ blocks is constructed. Then, a quantum circuit is generated by randomly masking a number of quantum gates, as shown in Fig.10 and Table.14. Finally, for each quantum circuit, 20 quantum operators are obtained by sampling quantum gates from the Haar-Gaussian distribution and contracting them according to the generated quantum circuit structure.

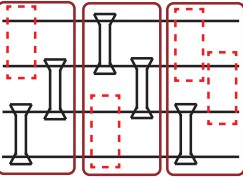

quantum circuit (4 qubits)

Figure 10: Example of the quantum circuits generated for the synthetic quantum operator experiment ($Q = 4$).

Table 14: TNO structure for generation of synthetic quantum operators.

| #qubits | #block | Mask locations for each block | | | | |
|---------|--------|---------|---------|---------|---------|---------|
| $Q$ | $M$ | Block 1 | Block 2 | Block 3 | Block 4 | Block 5 |
| 4 | 3 | 1 | 2 | 1, 3 | / | / |
| 8 | 5 | 3, 4, 6 | 2, 3, 6, 7 | 1, 2, 4, 5, 7 | 1, 3, 5, 6 | 2, 4, 5, 6 |
| 12 | 3 | 2, 3, 4, 6, 8, 10 | 1, 3, 5, 6, 7, 9, 11 | 1, 2, 4, 5, 7, 8, 9, 10, 11 | / | / |

## D.3 GENERATION OF QUANTUM FOURIER TRANSFORM (QFT) OPERATORS

The QFT is to represent the discrete Fourier transform (DFT) using a quantum circuit. As analyzed in (Chen et al., 2023; Camps et al., 2021), the DFT matrix can be efficiently represented by QFT using matrix (tensor) decomposition due to its special structure. In our experiment, we consider the DFT matrix $F$ of size $2^Q \times 2^Q$ with $Q = 4, 6$, and the QFT operator is set as $U = F$.

## D.4 ALGORITHM FOR QUANTUM CIRCUIT OPTIMIZATION

From the view of TNO, the quantum circuit can be seen as a TNO graph, and the quantum gates can be represented using the core tensors. Thus, representing a quantum operator $U$ is equivalent to approximating it using a TNO graph with some core tensors. Additionally, according to the property of quantum computing, we should constrain the core tensors to be unitary and relax the value to be complex-valued.

Concentrating on the goal of compact representation, we assume the $M$ for each generated synthetic quantum operator $U$ is known and directly deploy `Phase II` to the brick-wall TNO structure with $M$ blocks. While for each QFT operator, $M$ is set to a relatively large value to achieve a satisfactory high fidelity. At each evaluation, given a TNO and the quantum operator $U$, the density-matrix renormalization group (DMRG) method (Schollwöck, 2005) is applied to optimize the core tensors, with each core tensor initialized as an identity tensor. The loss function $\pi_D(\cdot)$ in Eq. (2) is defined as

$$\pi_D(X) = 1 - |tr(U^\dagger X)|/I^Q. \tag{7}$$

where $|tr(U^\dagger X)|/I^Q$ is the normalized fidelity quantifying the 'closeness' between two operators $U$ and $X$. The $\eta_2$ is set to $10^{-3}$. For each case of representing synthetic quantum operators, the fidelity is averaged over the generated 20 unitary operators.

## D.5 IMPLEMENTATION

Due to limited resources, we implement the quantum experiment just in the proof-of-concept scale using a laptop MacBook Air 13-inch with M3 chip and memory of 24 GB.

# E  PROOFS FOR PROPOSITION 3.1

In this section, we present the proof of Proposition 3.1 from the main manuscript. Here, we prove the explicit upper bound, naturally inducing the proposition provided in the main text.

**Proposition E.1** (Perturbation analysis). *Let $\mathcal{Z}^M = \{Z_1, \ldots, Z_M\}$ be a system of TNOs with $\mathcal{Z}^M \in \arg\min_{\mathcal{Y}^M \in ts(G_1, \ldots, G_M; Q, K, I)} \pi_D(\mathcal{Y}^M)$, where $\pi_D$ is a task loss defined jointly on multiple operators. Suppose one operator $Z_j$ contains a maskable core $\mathcal{G}$ with $\|\mathcal{G}\|_F = 1$ and operator–Schmidt rank $S$. Let $\mathcal{Z}^{M,\star} = \{Z_1, \ldots, Y_j^\star, \ldots, Z_M\}$ be the system obtained by replacing $Z_j$ with its best masked version*

$$Y_j^\star \in \arg\min_{Y \in tno(G_{\mathrm{mask}}; Q, K, I)} \pi_D(\{Z_1, \ldots, Y, \ldots, Z_M\}),$$

*where $G_{\mathrm{mask}}$ is obtained by masking $\mathcal{G}$. If $\pi_D$ is L-Lipschitz in $\|\cdot\|_F$ with respect to each operator, then*

$$|\pi_D(\mathcal{Z}^{M,\star}) - \pi_D(\mathcal{Z}^M)| \leq L\,C\,\sqrt{S-1},$$

*where $C$ depends only on the contracted environment of $\mathcal{G}$. In particular, if $S = 1$, masking $\mathcal{G}$ does not increase the loss.*

*Proof.* Let $\mathcal{Z}^M = \{Z_1, \ldots, Z_M\} \in ts(G_1, \ldots, G_M; Q, K, I)$ be system-optimal:

$$\mathcal{Z}^M \in \arg\min_{\mathcal{Y}^M \in ts(G_1, \ldots, G_M)} \pi_D(\mathcal{Y}^M).$$

Fix an index $j$ and a maskable core $\mathcal{G}$ inside $Z_j$ with $\|\mathcal{G}\|_F = 1$ and operator–Schmidt decomposition $\mathcal{G} = \sum_{r=1}^S \lambda_r \mathcal{U}_r \otimes \mathcal{V}_r$, where $\lambda_1 \geq \cdots \geq \lambda_S \geq 0$ and $\|\mathcal{U}_r\|_F = \|\mathcal{V}_r\|_F = 1$. By the bipartition assumption, contracting the entire network except $\mathcal{G}$ defines a linear map

$$\mathcal{T}: \mathbb{F}^{I^K \times I^K} \longrightarrow \mathbb{F}^{I^Q \times I^Q}, \qquad \text{such that } Z_j = \mathcal{T}(\mathcal{G}).$$

Since $\mathcal{T}$ is linear, write

$$Z_j = \mathcal{T}(\mathcal{G}) = \sum_{r=1}^S \lambda_r\,\mathcal{T}(\mathcal{U}_r \otimes \mathcal{V}_r).$$

**A realizable masked candidate.** Masking $\mathcal{G}$ removes that core and reconnects its neighbors. Because the top Schmidt term $\mathcal{U}_1 \otimes \mathcal{V}_1$ factorizes across the bipartition, it can be absorbed into the adjacent cores on each side of the cut, hence is realizable in the masked class. Therefore, there exists $Y_j^{(0)} \in tno(G_{\mathrm{mask}}; Q, K, I)$ such that

$$Y_j^{(0)} = \mathcal{T}(\mathcal{U}_1 \otimes \mathcal{V}_1).$$

Define the system candidate $\mathcal{Z}^{M,(0)} := \{Z_1, \ldots, Y_j^{(0)}, \ldots, Z_M\}$.

**Environment norm bound.** Let $C := \|\mathcal{T}\|_{2 \to F} := \sup_{X \neq 0} \frac{\|\mathcal{T}(X)\|_F}{\|X\|_F}$, which depends only on the (fixed) contracted environment surrounding $\mathcal{G}$. Then

$$\|Z_j - Y_j^{(0)}\|_F = \Big\|\sum_{r=2}^S \lambda_r\,\mathcal{T}(\mathcal{U}_r \otimes \mathcal{V}_r)\Big\|_F \leq \sum_{r=2}^S \lambda_r\,\|\mathcal{T}(\mathcal{U}_r \otimes \mathcal{V}_r)\|_F \leq C \sum_{r=2}^S \lambda_r.$$

Since $\|\mathcal{G}\|_F^2 = \sum_{r=1}^S \lambda_r^2 = 1$, Cauchy–Schwarz yields $\sum_{r=2}^S \lambda_r \leq \sqrt{S-1}\,(\sum_{r=2}^S \lambda_r^2)^{1/2} \leq \sqrt{S-1}$, hence

$$\|Z_j - Y_j^{(0)}\|_F \leq C\,\sqrt{S-1}.$$

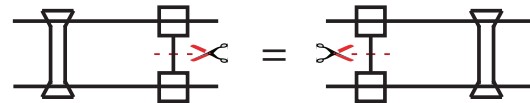

Figure 11: A simple example demonstrating the equivalence of masking different core tensors in a QTN.

**Lipschitz transfer to task loss.** By the $L$-Lipschitz property of $\pi_D$ w.r.t. each operator in Frobenius norm,

$$|\pi_D(\mathcal{Z}^{M,(0)}) - \pi_D(\mathcal{Z}^M)| \leq L\,\|Z_j - Y_j^{(0)}\|_F \leq L\,C\,\sqrt{S-1}.$$

**Optimal masked replacement.** By definition,

$$Y_j^\star \in \arg\min_{Y \in tno(G_{\text{mask}}; Q,K,I)} \pi_D(\{Z_1, \ldots, Y, \ldots, Z_M\}),$$

so $\pi_D(\mathcal{Z}^{M,\star}) \leq \pi_D(\mathcal{Z}^{M,(0)})$, where $\mathcal{Z}^{M,\star} := \{Z_1, \ldots, Y_j^\star, \ldots, Z_M\}$. Therefore,

$$|\pi_D(\mathcal{Z}^{M,\star}) - \pi_D(\mathcal{Z}^M)| \leq |\pi_D(\mathcal{Z}^{M,(0)}) - \pi_D(\mathcal{Z}^M)| \leq L\,C\,\sqrt{S-1}.$$

Finally, if $S = 1$ then $Z_j = \mathcal{T}(\mathcal{U}_1 \otimes \mathcal{V}_1)$ is itself realizable post-masking, so we may choose $Y_j^{(0)} = Z_j$, giving zero loss change. This proves the claim. $\qquad\square$

We emphasize that a small Schmidt rank, as discussed in Proposition 3.1, is *not* a necessary condition for masking a core tensor to have minimal impact on approximation error. This is demonstrated by a simple yet insightful example in Figure 11, where two core tensors occupy a commutative position within a TNO, and each individually possesses full model representation. In this case, masking either core tensor yields an equivalent result, regardless of its Schmidt number. More generally, this condition holds when the Zariski closure (Landsberg et al., 2012) of the remaining TNO spans the entire ambient space $\mathbb{F}^{I^K \times I^K}$. Such universal representation properties are well-studied in the context of quantum computing Barenco et al. (1995); Möttönen et al. (2004).

# F    STATEMENT OF THE USE OF LARGE LANGUAGE MODELS (LLMS)

During the writing of this paper, language polishing and grammar checking were partially assisted by Large Language Models (LLMs). The LLMs were used to improve the accuracy and fluency of the text, with all modifications reviewed and approved by the author to ensure originality and academic integrity.

