# OpenReview forum: "Joint Structure Search for Tensor Network Operators Inspired by Symmetry Breaking"
_ICLR.cc/2026/Conference — Submitted to ICLR 2026_

### Official Review · Reviewer_ADcS · 2025-10-21

**Soundness:** 3
**Presentation:** 1
**Contribution:** 2
**Rating:** 4
**Confidence:** 2

**Summary:**

The paper proposes a new tensor network operator system structure search approach. Inspired by symmetry breaking in physics, they propose a two phase optimization procedure: First, they run normal tensor networ structure search, then they add a regularizer in the structure search optimization problem, which encourages asymmetric task-specific tensor network structures. The regularizer takes the form of a simple core tensor masking. They show that this formulation yields significantly more compact tensor representations in three distinct tensor network settings: Tensor network decomposition, parameter-efficient fine-tuning, and quantum circuit optimization.

**Strengths:**

Tensor networks seem to be a general enough formulation that this might have a lot of use cases, although I am a bit unsure about it, see the weaknesses section

The results of the proposed algorithm look convincing, consistently yielding good performance.

**Weaknesses:**

I have two concerns with the current paper:

1.I find the paper quite inaccessible in its current form for readers not already familiar with tensor networks. A better motivation of tensor networks and the structure search setup, with an example formulation for an ML application, would be helpful at the beginning or at least in the appendix of the paper. Something of the form: A tensor network is an expression of the form einsum(A_ij,B_jk,C_kl), with A,B,C being called core tensors and could stand for ... in < application>.

The metaphor with the Higgs potential also seems unhelpful to me; I don't see how the Higgs potential maps to tensor networks or the structure search problem. In my opinion, as someone not very familiar with this topic, it did not aid my understanding, and the space would be better used for more intuitive motivation and problem setup of tensor network structure search in general, and how it can be useful for an ML practitioner.


2.The method is specifically designed for high-order tensor networks. While I have no doubt that they are common in computational physics, I am unsure how common these forms of tensor networks are in ML specifically. Could you give some more examples where these methods could be useful in ML?

PEFT for LLMs is given as an ML example, but also prefaced that it is not intended as a new practical PEFT method. Could you expand on what's missing for a practical algorithm?

**Questions:**

Instead of the proposed regularizer, could one just directly add the number of parameters into the optimization problem to incentivize more efficient solutions?

Overall, I am currently unsure about the paper, in particular the relevance for an ML conference, but if the questions are addressed satisfactorily, I am willing to increase my score.

---

> ### Author Response · Authors · 2025-11-14
> **Clarifying the Generality and ML Relevance of Tensor Networks**
>
> Dear Reviewer,
>
> Thank you very much for the evaluation of our work and for providing constructive, non-boilerplate comments:) Below, we respond to your concerns in the following posts. First of all, we would like to address your following concern:
>
> ---
>
> ### 1. *“Tensor networks seem to be a general enough formulation that this might have a lot of use cases, although I am a bit unsure about it.”*  and  *“The method is designed for high-order TNs; unsure how common these are in ML. More examples?”*
>
> **R:**  Thank you for the question. Tensor networks (TNs) are indeed a general and useful framework widely used to model high-dimensional linear operators. In the ML literature, TNs already appear in a broad range of applications, including model compression [1,2], LLM training and fine-tuning [3–5], reinforcement learning [6], prompt learning [7], statistical learning [8,9], physics-inspired ML [10] and many foundational studies on tensor decomposition, up to the ones already cited in our paper. These works routinely employ medium- to high-order TNs, fully aligned with the setting of our paper. Even higher-order TNs are common in quantum ML and scientific computing.
>
> Our work is directly motivated by these studies: while they demonstrate the effectiveness of TN-based modeling, they also highlight the challenge of *model selection*. The goal of our paper is to develop a more efficient model selection method, particularly in heterogeneous multi-operator settings, which is more practical and investigated detailedly for the first time.
>
> In the revision, we plan to enrich the overview of TN applications in ML at the very beginning to help readers, especially those less familiar with tensor networks, better understand their relevance across modern ML tasks.
>
> We hope this clarification helps address your concern. Thank you again for your thoughtful feedback.
>
> We will provide responses to your remaining points in the upcoming posts shortly.
>
> **References:**
>
> [1] Novikov, Alexander, et al. "Tensorizing neural networks." Neurips’15.
>
> [2]Kossaifi, Jean, et al. "Tensor regression networks." JMLR 21.123 (2020): 1-21.
>
> [3] Yang, Zi, et al. "Comera: Computing-and memory-efficient training via rank-adaptive tensor optimization." Neurips’24.
>
> [4] Tao, Zerui, et al. "Transformed Low-rank Adaptation via Tensor Decomposition and Its Applications to Text-to-image Models." ICCV’25.
>
> [5] Chen, Zhuo, et al. "Quanta: Efficient high-rank fine-tuning of llms with quantum-informed tensor adaptation." Neurips’24
>
> [6] Sozykin, Konstantin, et al. "TTOpt: A maximum volume quantized tensor train-based optimization and its application to reinforcement learning." Neurips’22.
>
> [7] Qiu, Yuning, et al. "STEPS: Sequential Probability Tensor Estimation for Text-to-Image Hard Prompt Search." CVPR’25.
>
> [8] Han, Rungang, Rebecca Willett, and Anru R. Zhang. "An optimal statistical and computational framework for generalized tensor estimation." Ann. Stat. 50.1 (2022): 1-29.
>
> [9]Saiapin, Albert, and Kim Batselier. "Tensor Network Based Feature Learning Model." AISTATS’25.
>
> [10] Stoudenmire, Edwin, and David J. Schwab. "Supervised learning with tensor networks." Neurips’16.

---

> ### Author Response · Authors · 2025-11-17
> **Paper revision for improving the accessibility**
>
> In this post, we would like to focus on the revisions we made to address your concern regarding the accessibility of the paper.
>
> ---
>
> ### 2. *“I find the paper quite inaccessible… better motivation of tensor networks and structure search needed.”*
>
> **R:**  We appreciate you this constructive suggestion! In the revision, we have made several substantial improvements to enhance accessibility for readers less familiar with tensor networks. Please also refer to the updated manuscript for the fully revised content. Below are the revision highlights:
>
> ---
>
> ### **(a) Added a new appendix section: “A Practical and Intuitive Introduction to Joint TN-SS”**
> At the beginning of the Appendix, we introduce a new section that re-explains the core ideas of tensor network operators (TNOs), tensor network systems, and tensor contractions in an _intuitive_ manner. Following your suggestion, we included simple einsum-based examples to illustrate TNO implementations in practice.
>
> The main text presents the concepts still in original form, for the purpose of  mathematical rigor, but we have revised the wording throughout to include more intuitive explanations and an inline einsum illustration where appropriate.
>
> ---
>
> ### **(b) Clarified the TN and TN-SS tasks with explicit ML connections**
> Apart from our earlier response regarding the relevance of TNs in ML, we have also rewritten the first part of the Related Work section, changing the content from  *“Tensor networks (TNs) in representing linear operators”*  to  *“Tensor networks in machine learning”*.
>
> This update better reflects the practical ML contexts where TNs appear.
> We also revised the introduction section to provide a clearer motivation for joint TN-SS and its relevance for ML practitioners.
>
> ---
>
> ### **(c) Replaced and relocated the Higgs potential metaphor**
> Thank you for pointing this out. In the revised version, we removed the original section of _“A BRIEF REVIEW OF SYMMETRY BREAKING AND CONNECTION TO JOINT TN-SS”_.
>
> To preserve the underlying intuition without relying on physics-specific metaphors, we replaced that section with a concise and more accessible explanation. In the new pargraphs, we hightlight the main intuition into the following secentence:
>
> > *The optimal solution to a difficult high-dimensional problem often lies close to a suboptimal solution obtained in a simpler, lower-dimensional search space.*
>
> We added a short paragraph explaining how this intuition directly motivates our two-phase design:
> Phase I performs a coarse search in a simplified structure space (i.e., symmetry), and Phase II then refines the search by breaking symmetry and exploring asymmetric task-specific structures.
>
> We also provide additional clarification in the appendix for ML readers who may not be familiar with symmetry-breaking concepts.
>
> ---
>
> We hope these revisions make the paper more accessible and intuitive for general ML readers, while preserving technical rigor.
>
> Best regards,
> the authors

---

> > ### Author Response · Authors · 2025-11-17
> >
> > In this post, we would like to respond to the following two questions:
> >
> > ---
> >
> > ### 3. *“PEFT example is not intended as a practical PEFT method. What is missing for a practical algorithm?”*
> >
> > **R:**  Thank you for this question, and for noting our careful statement in the paper regarding the PEFT experiments:
> >
> > > *“… this experiment is not intended to propose a new practical PEFT method, but rather to highlight how joint TN-SS itself contributes to parameter-efficient representations.”*
> >
> > We intentionally set a limited goal for the PEFT experiment, namely, to demonstrate how joint TN-SS improves parameter efficiency when applied on top of an existing TNO-based PEFT approach. This choice avoids overstating the contribution as a full-fledged practical PEFT pipeline.
> >
> > If the goal were instead to build a *practical* PEFT method, additional investigations would be required, including:
> >
> > - Training on much larger models (e.g., 70B-scale LLMs or beyond);
> > - Evaluating training difficulty and stability across multiple datasets and tasks;
> > - Analyzing real computational and memory costs instead of only parameter count;
> > - Demonstrating plug-in compatibility with existing PEFT methods;
> > - Comparing with a broader set of recent baselines.
> >
> > These studies would certainly strengthen the contribution, but they extend beyond the current scope, which focuses on showing that joint TN-SS can automatically discover more compact and expressive TNO-based structures, an ability that naturally benefits parameter-efficient learning such as PEFT.
> >
> > We also note that our work is the *first* to connect TN-SS to PEFT. A more comprehensive exploration toward a practical PEFT framework built upon joint TN-SS is an exciting direction we plan to pursue in future work.
> >
> > ---
> >
> > ### 4. *“Could one directly regularize the number of parameters instead of the proposed regularizer?”*
> >
> > **R:**  We appreciate this insightful question. The answer is yes. Directly penalizing the number of parameters has been explored in earlier TN-SS works (e.g., Li et al., ICML ’20, ’22) for conventional tensor networks without circuit-format constraints.
> >
> > However, in the context of _joint TN-SS on TNOs_, parameter-count penalization does *not* naturally induce an implementable algorithm that preserves the required circuit-like structural constraints. In contrast, our masking-based regularizer operates directly on core tensors, ensuring that every pruned configuration remains a *valid TNO operator* throughout the search.
> >
> > Moreover, our formulation introduces only two *task-transparent* hyperparameters $(\eta_1, \eta_2)$. In contrast, parameter-penalization–based methods might impose explicit weighting coefficients in the objective function, which often reduces interpretability and complicates tuning.
> >
> > We will clarify this design choice in the revision to avoid confusion.
> >
> > ---
> >
> > Thank you again for your time in reading our response. Please let us know if our explanation addresses your concerns. Any further discussion is very welcome and appreciated.
> >
> > Best regards,
> > The authors

---

> > > ### Comment · Reviewer_ADcS · 2025-11-21
> > >
> > > Thank you for the thorough response. The revised PDF does seem to give a much more thorough introduction and seems overall more accessible. From my point of view, the paper seems to be in good shape with a solid contribution that, while slightly niche, might be of interest to a non-negligible subset of the ML community. Therefore, I recommend the paper for acceptance.

---

> > > > ### Author Response · Authors · 2025-11-26
> > > >
> > > > Thanks a lot for your positive response and increasing the score!

---

### Official Review · Reviewer_rZHz · 2025-10-28

**Soundness:** 3
**Presentation:** 4
**Contribution:** 2
**Rating:** 4
**Confidence:** 2

**Summary:**

This paper introduces a framework for joint tensor network structure search (joint TN SS) in systems that involve multiple linear operators, such as neural network layers or quantum circuit blocks. The authors propose a two-stage algorithm inspired by the concept of symmetry breaking in physics:
- Phase I (symmetry stage): All operators are constrained to share a common tensor network structure. A standard TN SS method is then applied to identify an expressive shared graph G.
- Phase II (symmetry-breaking stage): Operator-specific variations are introduced through a greedy core-masking process, which removes selected cores as long as the overall loss remains within a user-defined tolerance η_2. The transition between phases is controlled by two tolerance parameters η_1and η_2, reflecting a mechanism similar to the Mexican-hat potential in physics.

The approach is evaluated on three tasks: synthetic tensor decomposition, parameter-efficient fine-tuning of large language models (using the QuanTA method on Llama2 7B), and quantum circuit optimization (including QFT). Across these domains, the method achieves comparable or better accuracy while reducing the number of cores or parameters and lowering search costs compared to existing TN SS baselines such as TNGA, TNLS, TnALE, and Greedy.

**Strengths:**

- Well-defined problem and strong motivation: The paper clearly addresses the challenge of handling heterogeneous operator structures in multi-operator systems and formalizes this through TNO systems and joint structure search.
- Straightforward and general approach: The proposed two-phase method—starting with shared structure and then introducing diversity—is conceptually simple, easy to implement, and builds on existing TN-SS techniques. The use of core masking in Phase II offers an efficient way to add variability at low cost.
- Interpretability of design choices: Linking the phase transition to task-specific tolerances (η_1,η_2) makes the method more transparent compared to approaches that rely on opaque regularization parameters.
- Comprehensive experimental coverage: The evaluation spans synthetic tensor decomposition, parameter-efficient fine-tuning of LLMs (Llama2-7B across multiple reasoning benchmarks), and quantum circuit optimization, demonstrating the method’s versatility.
- Theoretical contribution: Proposition 3.1 provides a useful criterion for pruning based on operator–Schmidt rank, which is relevant for understanding structural efficiency in tensor networks.

**Weaknesses:**

- Dependence on a shared graph: The method assumes that a sufficiently expressive common graph Gexists, drawing on the universal approximation property of circuit-like TNs. In practice, selecting Gand its order Lis challenging, and the paper acknowledges failure cases when Lis mis-specified without offering a clear solution. A systematic way to choose Lwould make the approach more robust.
- Phase II is heuristic: The diversification step relies on greedy core masking, which may be sensitive to the choice of cores and can get stuck in suboptimal configurations. Other diversification strategies—such as core splitting, rank adjustment, or permutation changes—are not explored.
- Limited scope in PEFT experiments: While the PEFT section is presented as illustrative, the comparisons are narrow. Stronger baselines like rank-adaptive methods (e.g., CoMERA, GaLore/WeLore) and adapter mixtures (e.g., MixLoRA, DoRA variants) are missing, which limits the strength of the claims.
- Small-scale quantum results: The quantum circuit experiments are restricted to modest sizes (synthetic cases up to Q=12, QFT up to Q=6) and use basic DMRG setups on limited hardware. This leaves open questions about scalability and performance in realistic scenarios.
- Theory lacks practical guidance: Proposition 3.1 depends on constants such as the environment norm C and Lipschitz parameter L, but these are not estimated or validated empirically. The bound may be loose, and the paper does not provide guidance on how to compute or approximate these values in practice.
- Reproducibility concerns: Code for quantum circuit optimization is not yet available and is promised only after publication. Given the importance of that section, early release would improve transparency and allow verification.

**Questions:**

N/A

---

> ### Author Response · Authors · 2025-11-13
>
> Dear Reviewer,
>
> Thank you for providing your review. Below, we address each concern raised in the *Weakness* section and clarify how we will further improve the paper.
>
> ---
>
> ### **1. On selecting \(L\): _“A systematic way to choose \(L\) would make the approach more robust.”_**
>
> **R:**  We appreciate this question. Appendix A.6 (Fig. 6) already provides an empirical sensitivity analysis of \(L\) and explains why it influences performance. In practice, we propose a simple and robust selection rule:
>
> -  Treat \(L\) as a hyperparameter guided by the Phase-I tolerance \(\eta_1\) (which you noted is task-transparent).
> -   Start from a small, computationally inexpensive \(L\), and gradually increase it until the desired \(\eta_1\) is achieved.
>
> In the revision, we will move this guideline to the main text as a clear rule-of-thumb recipe.
>
> We also note respectfully that tuning \(L\) in this way is no more difficult than tuning standard ML hyperparameters such as network depth or hidden width.
>
> ---
>
> ### **2. _“Phase II is heuristic… may be sensitive/suboptimal; other diversification strategies not explored.”_**
>
> **R:**  We respectfully disagree that being heuristic is a weakness specific to our approach. TN-SS is NP-hard (explicitly stated in the Introduction), and *all* practical TN-SS methods necessarily rely on heuristic strategies.
>
> Our contribution is to introduce a principled symmetry-breaking framework, and we show that even a simple greedy masking mechanism works consistently well across all tested domains.
>
> The method’s stability and robustness are demonstrated through extensive experiments in both the main text and Appendix. These results consistently show superior or comparable performance to strong baselines.
>
> While we agree that diversification strategies such as core-splitting, rank adjustment, and permutation search are interesting directions, they are temporally out of the scope of this work. We will explicitly mention them as natural future extensions.
>
> ---
>
> ### **3. _“Limited scope in PEFT experiments; missing stronger baselines.”_**
>
> **R:**  Thank you for raising this point. To clarify:
>
> -  Rank-adaptive methods (CoMERA, GaLore/WeLore, where CoMERA and GaLore are already cited) focus primarily on low-rank efficient training. While WeLore includes PEFT elements, these methods target objectives different from structure search, which is our focus.
> - For adapter mixtures and LoRA variants, we already include LoRA, DoRA, and QuanTA (NeurIPS’24). QuanTA is a strong tensor-based PEFT method on which our approach builds.
>
> Our PEFT experiments are deliberately scoped:
>
> > The goal is to demonstrate how *joint TN-SS improves parameter efficiency on top of an existing TNO-based PEFT method*, rather than to exhaustively benchmark all PEFT algorithms.
>
> We have taken care to avoid overstating claims. In the revision, we will further emphasize this positioning and add discussion showing that our method is complementary and can be integrated with rank-adaptive or mixture-based methods.
>
> ---
>
> ### **4. _“Theory lacks practical guidance; constants \(C\) and \(L\) not estimated.”_**
>
> **R:**  The purpose of Proposition 3.1 is to identify the key factor governing loss sensitivity, namely, the operator–Schmidt rank of the core being masked. This theoretical insight directly informs Phase II’s design. In the bound:
>
> -  The Lipschitz constant \(L\) is a *standard assumption* in sensitivity analysis.
> -  The environment constant \(C\) is a *task-dependent scaling factor*, and not meant to provide additional insightful information.
>
> In the revision, we will clarify that the bound is intended qualitatively, for example emphasizing the \(\mathcal{O}(S^{1/2})\) dependence, rather than providing exact bounds.
>
> ---
>
> ### **Final Remarks**
>
> We thank you again for the feedback.  The clarifications above, together with the planned textual improvements, will strengthen the clarity, precision, and scope of the paper. Additional responses and updates will follow.
>
> We welcome and greatly appreciate any further discussion.
>
> Best regards,
> the authors

---

> ### Author Response · Authors · 2025-11-13
>
> ### **5. “_Small-scale quantum results; unclear scalability._”**
>
> **R:**  We fully agree that the current quantum experiments are modest in scale (as explicitly acknowledged in the Limitation section, and the scale has been larger than experiments in *all* existing TN-SS studies.). For clarification, we would like to first highlight that, like most TN-SS methods, joint TN-SS is also a  meta-algorithm, meaning that its _overall scalability depends on a task-dependent inner optimization procedure, which lies out of the scope of our work_. The quantum-circuit experiments are currently limited to modest scales due to the memory cost of such the specific inner optimization we implemented. Furthermore, we also believe the present scale is sufficient to support the key claim of this work:
>
> > joint TN-SS consistently discovers significantly more compact (in some cases previously unknown) circuit structures compared with widely used baselines.
>
> This is precisely what we observe in both the synthetic data and the QFT experiment.
>
> To avoid any misunderstanding, we respectfully re-emphasize that the scale limitations are *already clearly stated* in the paper:
>
> - Sec. 5 explicitly states that the quantum experiments are small-scale and that scaling to larger systems is future work.
> - Appendix C.5 explains that the experiments were run on a laptop, which restricted the maximum qubit size.
>
> In the revision, we will make this positioning even more explicit: the quantum results are intended as proof-of-concept demonstrations validating the framework.
>
> ---
>
> ### **6. “_Reproducibility concerns: quantum code only after publication._”**
>
> **R:**  We agree that early code release improves transparency. In our paper,
>
> - For tensor decomposition and PEFT, all code and detailed configurations are already provided with the initial submission.
> - For the quantum circuit decomposition experiments, we now additionally provide an anonymous reproducible implementation, enabling complete verification of the QCD results (Sorry for the delay, due to privacy sanitization of internal  codes and metadata):
>
> **Anonymous code link:**
> https://anonymous.4open.science/r/test-927F
>
> Any futher disucsion on these points are welcome and appreciated.

---

> > ### Comment · Reviewer_rZHz · 2025-11-24
> > **Response to rebuttal**
> >
> > Dear Authors:
> >
> > Thank you for your response and clarifications. However, although you responded to me in a detailed manner, most of your explanations were simply along the lines of "xxx is standard" and "xxx is meant to be yyy". Basically it was a reiteration of what was previously stated in the paper, without actually providing too much new information.
> >
> > I still fail to see the major benefits of this newly proposed framework but I want to remind the AC that this is not my research field and I have rather limited knowledge on it, so please take it with a grain of salt.

---

> > > ### Author Response · Authors · 2025-11-30
> > >
> > > Thank you for the direct feedback. Regarding your concern that our earlier explanation sounded like “this is standard” or “this is meant to be,” (we assume this mainly refers to Proposition 3.1), we agree that our previous wording may have felt too general. A few of the initial comments we received were phrased at a high level or in a somewhat generic style, and we now realize that our response could have provided more intuitive clarification, especially given that this area sits at a rather specialized intersection of tensor networks, structure search, quantum computation, and ML. We are very happy to offer a clearer explanation below.
> > >
> > > To clarify more for Prop. 3.1: its purpose is not to provide numerically precise constants, but to isolate the key structural factor that determines loss sensitivity in Phase II, namely, the operator–Schmidt rank $S$ of the masked core. The constants $L$ (a Lipschitz bound) and $C$ (an environment scaling factor) are intentionally not estimated because, as in most perturbation analyses in ML, they depend on task-specific scale and loss function and are not instructive to practitioners.
> > >
> > > What the proposition does tell us is that, up to these problem-dependent constants, the dominant term governing the loss change is $\mathcal{O}(\sqrt{S-1})$ (**Revision:** we have carefully revised it in the main text). This is the theory to say when masking low-Schmidt-rank cores is safe and why Phase II is structured the way it is.
> > >
> > >
> > > **Highlight of major benefits.** Because this topic spans multiple specialized areas, the broader value of the framework may not be immediately apparent without domain familiarity. In more accessible terms: the key benefit is that our framework is the first to provide a *unified and efficient method* for automatically discovering compact structures for multiple related operators: a setting that naturally arises in many modern ML pipelines (e.g., different LLM layers, tensorized compression modules, multi-operator quantum circuits) but is not handled by existing TN-SS methods. In practice, this results in direct and measurable benefits: fewer parameters, more efficient tensor representations, and substantially more compact quantum circuits, all obtained automatically without extensive hand-crafted architecture design. Conceptually, the symmetry-breaking mechanism offers a practical coarse-to-fine search principle that we believe may inspire future work in neural architecture search, combinatorial optimization, and other discrete structure-learning problems.
> > >
> > > We appreciate your engagement with our work. Your feedback has been valuable in helping us improve the clarity and positioning of the paper.

---

> > > > ### Author Response · Authors · 2025-11-30
> > > > **Updated PEFT Results Incorporating New Baselines**
> > > >
> > > > Dear Reviewer,
> > > >
> > > > We have conducted new PEFT experiments with several stronger and more diverse baselines (emphasized with Italic font), including MixLoRA, MixDoRA (Li et al., 2024a), VB-LoRA (Li et al., 2024b), and VeRA (Kopiczko et al., 2023). The updated results (see the table below) show that _our jointly searched TNO structures continue to outperform these baselines at matched or lower parameter budgets_. This strengthens our claim that joint TN-SS offers a parameter-efficient and competitive PEFT solution even compared against recent state-of-the-art methods. The result is also updated in the main text.
> > > >
> > > > Best regards,
> > > >
> > > > ---
> > > >
> > > > | Method      | \#Params | PIQA | SIQA | OBQA | ARC-e | ARC-c | Avg. |
> > > > |-------------|:--------:|:----:|:----:|:----:|:-----:|:-----:|:----:|
> > > > | LoRA    |  3.200%  | 82.1 | 69.9 | 80.4 |  73.8 |  50.9 | 71.4 |
> > > > | DoRA    |  3.200%  | 82.7 | 74.1 | 80.6 |  76.5 |  59.8 | 74.7 |
> > > > | *MixLoRA* |  *3.200%*  | *83.2* | *78.0* | *81.6* |  *77.7* |  *58.1* | *75.7* |
> > > > | *MixDoRA* |  *3.200%*  | *82.2* | *80.4* | *80.9* |  *77.5* |  *58.2* | *75.8* |
> > > > |-------------|       |        |         |      |        |        |        |
> > > > | *VB-LoRA*     |  *0.042%*  | *74.8* | *74.9* | *78.0* |  *81.7* |  *60.8* | *74.0* |
> > > > | QuanTA-6    |  0.041%  | 79.9 | 75.9 | 80.0 |  84.8 |  63.3 | 76.8 |
> > > > | **Ours** |  **0.031%**  | 80.4 | 76.2 | 80.2 |  84.7 |  63.7 | **77.0** |
> > > > |-------------|       |        |         |      |        |        |        |
> > > > | QuanTA-4    |  0.024%  | 79.2 | 73.5 | 77.2 |  84.4 |  62.6 | 75.4 |
> > > > | **Ours** |  **0.024%**  | 80.3 | 75.1 | 78.4 |  84.2 |  63.4 | **76.3** |
> > > > |-------------|       |        |         |      |        |        |        |
> > > > | *VeRA*        |  *0.018%*  | *77.5* | *68.3* | *72.4* |  *79.3* |  *53.7* | *70.2* |
> > > > | QuanTA-2    |  0.017%  | 78.1 | 72.5 | 74.8 |  82.1 |  57.1 | 72.9 |
> > > > | **Ours** |  **0.017%**  | 79.7 | 72.7 | 78.0 |  83.1 |  59.9 | **74.7** |

---

### Official Review · Reviewer_AEK4 · 2025-11-01

**Soundness:** 4
**Presentation:** 3
**Contribution:** 3
**Rating:** 8
**Confidence:** 5

**Summary:**

This paper introduces joint tensor network structure search (joint TN-SS) for systems with multiple linear operators (e.g., multi-layer Transformers, quantum circuits). Inspired by symmetry breaking in physics, the method balances a shared structure with operator-specific diversity to obtain more compact and accurate tensor network operator (TNO) representations. The algorithm proceeds in two phases: (i) symmetry phase, where a standard TN-SS finds a common structure expressive enough for all operators; (ii) symmetry-breaking phase, where greedy core masking introduces operator-specific specialization, controlled by task-explainable loss tolerances (η1, η2). The paper formalizes circuit-like TNOs, proposes a vertex-indexed incidence (VI) matrix to encode structures, and recasts joint TN-SS with an objective whose “negative regularization” plays the role of a mass parameter to trigger diversification. Experiments across synthetic tensor decomposition, parameter-efficient fine-tuning (PEFT) for LLMs, and quantum circuit optimization show more compact structures with equal or better accuracy than baselines at comparable or lower search cost. A perturbation bound further shows low operator–Schmidt rank cores can be masked with no or bounded loss increase.

**Strengths:**

(1) Novelty in extending TN-SS from a single operator to multi-operator systems with a principled “symmetry to symmetry breaking” transition, yielding a simple, general, and scalable search paradigm.

(2) Clear method design: a formal treatment of circuit-like TNOs, VI-matrix encoding, and a unified objective; core masking as a low-cost structural perturbation; and tolerance thresholds replacing opaque regularization weights to improve explainability and usability.

(3) Broad empirical coverage: in synthetic tensor decomposition, the method substantially reduces core counts while maintaining RSE less than 1e−6; in LLM PEFT, it outperforms QuanTA variants and LoRA/DoRA with the same or fewer parameters; in quantum circuits, it achieves more compact representations than brick-wall baselines at high fidelity and matches classic QFT designs while avoiding SWAP gates.

(4) Efficiency: the two-phase search decomposes an exponential space into a shared-then-specialize process with evaluation cost growing roughly linearly in the number of operators in Phase II; it is markedly more efficient than naïvely composing all structures into one large graph.

(5) Theoretical support: a perturbation bound tied to operator–Schmidt rank offers a sufficient condition for safe masking and clarifies when accuracy is preserved, along with discussion on non-necessity.

**Weaknesses:**

(1) Quantum-circuit experiments are small-scale and primarily proof-of-concept; scalability to larger Q, richer gate sets, and noisy hardware constraints remains to be demonstrated.

(2) Phase II uses greedy masking: fast and straightforward but potentially suboptimal. There is no systematic comparison with stronger global or metaheuristic searches (e.g., Bayesian optimization), nor an exploration of hybrid strategies.

(3) Tolerance choices ($\eta 1$/$\eta 2$) affect outcomes. While sensitivity analyses are given, there is no adaptive or learned policy for setting them across tasks, which may hinder plug-and-play deployment.

(4) Assumptions in TNO/VI encoding impose specific constraints on dimensions and graph structure; robustness and expressivity under heterogeneous dimensions and tightly coupled cross-layer patterns in real models are not fully assessed.

**Questions:**

(1) Can Phase II’s greedy masking be combined with stronger global heuristics (e.g., Bayesian optimization or gradient/sensitivity-informed prioritization) to improve optimality under tight budgets? How does accuracy–cost trade off?

(2) Can you propose and evaluate adaptive strategies for $\eta 1$/$\eta 2$ (e.g., validation-curve-based adjustment, early-stopping signals, or a learned threshold predictor) to reduce manual tuning and stabilize performance across tasks?

(3) Do you have generalized perturbation results for simultaneous multi-core masking and inter-operator coupling? Could spectral properties of the environment tensors yield computable criteria for safe pruning?

(4) On larger quantum circuits (bigger Q, constrained connectivity, noise models), and with hardware constraints (depth, two-qubit count, SWAP cost), how do benefits compare to heuristic synthesis/optimization tools?

(5) For LLM PEFT, what happens if search is more tightly coupled to training (e.g., few-step updates after each masking decision) to better reflect downstream generalization? Can compute be controlled via low-fidelity proxies?

---

> ### Author Response · Authors · 2025-11-20
> **Official Comment by Authors**
>
> Dear Reviewer,
>
> We sincerely appreciate your thoughtful feedback and constructive suggestions. Below, we address the concerns raised in the Weaknesses and Questions sections, and outline how we plan to further clarify and improve the manuscript.
>
> ---
>
> ### W1: *"Quantum-circuit experiments are small-scale and primarily proof-of-concept; scalability to larger Q, richer gate sets, and noisy hardware constraints remain to be demonstrated."*
>
> **R:** We fully acknowledge that the quantum-circuit experiments presented in this work are modest in scale, primarily serving as proof-of-concept. Nevertheless, we believe that the current experimental scale is sufficient to substantiate our central claim: joint TN-SS consistently discovers significantly more compact (and in some cases previously unknown) circuit structures compared to widely used baselines. This is clearly demonstrated in both synthetic benchmarks and the QFT experiment. Scaling joint TN-SS to larger qubit systems, richer gate sets, and realistic hardware constraints is an exciting avenue for future research. To avoid any misunderstanding, we respectfully emphasize that these limitations are already stated in the manuscript:
> - Section 5 explicitly notes that the quantum experiments are small-scale and that scaling to larger systems is future work.
> - Appendix C.5 clarifies that experiments were run on a laptop, which restricted the maximum qubit size.
>
> ---
>
> ### W2&Q1: *"Phase II uses greedy masking: fast and straightforward but potentially suboptimal. There is no systematic comparison with stronger global or metaheuristic searches (e.g., Bayesian optimization), nor an exploration of hybrid strategies. Can Phase II’s greedy masking be combined with stronger global heuristics (e.g., Bayesian optimization or gradient/sensitivity-informed prioritization) to improve optimality under tight budgets? How does accuracy–cost trade off?"*
>
> **R:** We agree that integrating global or metaheuristic search methods could further enhance the efficiency and optimality of Phase II. However, effectively quantifying the confidence for masking core tensors may increase algorithmic complexity, and developing such heuristics remains an open challenge in the TN-SS literature—including for joint TN-SS. We will explicitly highlight these issues as natural future directions. Our proposed method is the first attempt to address the joint TN-SS problem, and the greedy masking strategy has demonstrated desirable performance, validating the feasibility of solving this challenging task. We see substantial opportunities for improvement, as suggested, and we believe this work opens a promising new direction for future research.
>
> ---
>
> ### W3&Q2: *"Tolerance choices ($\eta_1$/$\eta_2$) affect outcomes. While sensitivity analyses are given, there is no adaptive or learned policy for setting them across tasks, which may hinder plug-and-play deployment. Can you propose and evaluate adaptive strategies for $\eta_1$/$\eta_2$ (e.g., validation-curve-based adjustment, early-stopping signals, or a learned threshold predictor) to reduce manual tuning and stabilize performance across tasks?"*
>
> **R:** We agree that adaptive strategies could improve the stability and usability of our method. However, as previously discussed, evaluation in TN-SS is challenging, and developing adaptive policies is especially difficult for greedy random core masking, where the loss trend is not always predictable. In the current work, we employ a simple yet effective strategy: $\eta_2$ is set slightly larger than $\eta_1$. Extensive numerical experiments across diverse tasks—including tensor decomposition, PEFT, and quantum circuits—demonstrate that this approach is stable and consistently outperforms strong baselines. More sophisticated adaptive strategies are an interesting direction for future work.
>
> We also emphasize that $\eta_1$ and $\eta_2$ are task-transparent and should be tuned according to the specific requirements of the application. As shown in the sensitivity analysis (Table 9), there is a tradeoff between performance and the choice of $\eta_2$ in Phase II: a larger $\eta_2$ yields higher accuracy with more core tensors, while a smaller $\eta_2$ results in a more compact representation with lower accuracy. For a given application, these parameters should be tuned to match the desired outcome.

---

> ### Author Response · Authors · 2025-11-20
> **Official Comment by Authors**
>
> Dear Reviewer,
>
> Below, we continue with our responses to the remaining weaknesses and questions raised in your review.
>
> ---
>
> ### W4: *"Assumptions in TNO/VI encoding impose specific constraints on dimensions and graph structure; robustness and expressivity under heterogeneous dimensions and tightly coupled cross-layer patterns in real models are not fully assessed. "*
>
> **R:**  The TNO structure provides a simple yet efficient TN representation, demonstrating promising applicability for PEFT and quantum operator optimization. Importantly, the proposed method can be extended to TNs with arbitrary graph structures, as the algorithms utilized in Phases I and II are not limited to the TNO structure. We will clarify this point in the revision to further emphasize the generality and extensibility of our approach.
>
> ---
>
> ### Q3: *"Do you have generalized perturbation results for simultaneous multi-core masking and inter-operator coupling? Could spectral properties of the environment tensors yield computable criteria for safe pruning?"*
>
> **R:**  Simultaneous multi-core masking can be treated as sequential masking—masking core tensors one-by-one. Therefore, it is straightforward to extend the proposed perturbation results to multi-core cases. The joint TN-SS algorithm uses loss as the criterion for evaluation and pruning, consistent with existing TN-SS approaches. Typically, there are no guaranteed spectral properties that can be leveraged to guide the pruning process, as TN-SS is inherently NP-hard. Developing spectral-based criteria for safe pruning lies beyond the scope of this work but represents an intriguing direction for future research.
>
> ---
>
> ### Q4: *"On larger quantum circuits (bigger Q, constrained connectivity, noise models), and with hardware constraints (depth, two-qubit count, SWAP cost), how do benefits compare to heuristic synthesis/optimization tools?"*
>
> **R:**  Compared to traditional approaches that optimize quantum circuits by iteratively applying local rules—such as gate elimination, gate merging, adjacent gate swaps, and greedy selection—within a vast search space, our proposed algorithm first enforces a globally shared structure (e.g., brick-wall topology). This is followed by symmetry breaking, where operator-specific diversity is introduced in a controlled manner. This joint TN-SS method releases structural diversity in an orderly fashion, effectively avoiding the combinatorial explosion associated with unconstrained local search. As a result, our method achieves structural compression while maintaining high fidelity. Notably, it not only reduces circuit depth but also mitigates the need for SWAP gates, thereby preventing the introduction of excessive tensor contractions that would otherwise increase quantum noise.
>
> ---
>
> ### Q5: *"For LLM PEFT, what happens if search is more tightly coupled to training (e.g., few-step updates after each masking decision) to better reflect downstream generalization? Can compute be controlled via low-fidelity proxies?”*
>
> **R:** We agree that more tightly coupling structure search with training could potentially improve the alignment between the searched structure and downstream generalization performance. This dynamic interaction allows the search algorithm to better capture the interplay between structural choices and optimization trajectories, which may be especially beneficial for PEFT scenarios where generalization is sensitive to fine-grained structural changes.
>
> However, this tighter coupling inevitably increases computational cost, as it requires repeated partial training or evaluation steps within the search loop. To address this, it is indeed feasible to control compute via low-fidelity proxies, such as short-horizon training, proxy metrics (e.g., gradient norms, parameter sparsity), or surrogate models. As the goal of the current work is to highlight how joint TN-SS itself contributes to parameter-efficient representations, integrating such proxies into our framework is beyond the scope of this work but represents a promising direction for future research toward more adaptive and scalable PEFT structure search.
>
> ---
>
> We greatly appreciate your constructive comments and suggestions. We welcome any further discussion and feedback.
>
> Best regards, the authors

---

> > ### Comment · Reviewer_AEK4 · 2025-11-25
> >
> > Thank the authors for the responses. I’m now satisfied that the current quantum experiments make sense to support the statements, and the tolerances parameters are easy to set in practice. My other concerns and questions are also well addressed.
> >
> > I encourage the authors to incorporate these clarifications into the final version for improved clarity and usability.
> >
> > Overall, I like the proposed idea. It’s simple, effective and well-motivated. The method shows potential for addressing computational problem for large models. Its conceptual connection to physics is also a strength. Therefore, I gave a good score for the acceptance recommendation.

---

> > > ### Author Response · Authors · 2025-11-26
> > >
> > > Thank you very much for your support and positive assessment!

---

### Official Review · Reviewer_TTpc · 2025-11-01

**Soundness:** 3
**Presentation:** 3
**Contribution:** 2
**Rating:** 4
**Confidence:** 3

**Summary:**

The authors present joint Tensor Network Structure Search, a new framework for discovering optimal tensor network structures when dealing with multiple coupled operators a setting common in large models like transformers or quantum circuits. The method operates in two phases: a symmetry phase, where a shared TN structure captures common inductive biases, and a symmetry-breaking phase, where operator-specific variations emerge through a greedy masking process guided by task-specific loss tolerances. This approach allows for both shared efficiency and individualized expressivity. Experiments across tensor decomposition, LLM fine-tuning, and quantum circuit optimization show that joint TN-SS achieves more compact models with equal or better accuracy than existing methods, all at a reasonable search cost.

**Strengths:**

S1. The paper introduces a novel extension of TN structure search to multi-operator systems, providing a fresh symmetry-based perspective that effectively balances shared structure and task-specific flexibility. The symmetry breaking concept is very interesting.

S2. The experimental validation spans diverse domains, from classical tensor tasks to quantum circuits, showing both versatility and efficiency gains.

S3. Proper limitations acknowledgments and related works section.

S4. The authors evaluate the proposed algorithm on diverse domains: joint tensor decomposition, parameter efficient fine-tuning of LLMs, and quantum circuit optimization with improvement results.

**Weaknesses:**

W1. The experiments on quantum circuits remain small-scale, so it’s unclear how the method performs under more realistic, large-system conditions.

W2. The optimization stability issues mentioned (e.g. with gradient or SVD-based methods) highlight that the current approach may still face robustness challenges in practical deployments.

W3. The significance to the broader ICLR community is not so clear as it the method is specific to Tensor Network Operators.

W4. The novelty is limited as it borrows on a elegant but relatively simple concept of symmetry breaking.

**Questions:**

Q1. How sensitive is the performance of joint TN-SS to the balance between the shared symmetry phase and the symmetry-breaking phase, does tuning this trade-off require manual effort?

Q2. Efficiency is mentioned 21 times but can you give some quantitative estimate on the efficiency gains expected within the overall framework proposed here?

Q3. Have you explored alternative optimization schemes to improve robustness, is there an ablation study?

Q4. How transferable are the discovered TN structures across different but related tasks, can a structure found for one operator family generalize to another without retraining?

---

> ### Author Response · Authors · 2025-11-18
>
> Dear Reviewer,
>
> Thank you  for taking the time to review our paper.  Due to the word limit, we respond here to the first two weaknesses you raised in this post.  We hope the clarifications below can address your concerns. Responses to the remaining points will follow in the next posts.
>
> ---
>
> ### W1. *The quantum circuit experiments remain small-scale, so it’s unclear how the method performs under more realistic, large-system conditions.*
>
> **R:**  We agree that the current quantum experiments are modest in scale (as explicitly acknowledged in the Limitations section). We also appreciate that you recognized our limitation discussion as a strength (S3).
>
> For clarification, we would like to highlight that, like most TN-SS methods, joint TN-SS is also a  meta-algorithm, meaning that its overall scalability depends on a task-dependent inner optimization procedure, which lies out of the scope of our work. The quantum-circuit experiments are currently limited to modest scales due to the memory cost of such the specific inner optimization we implemented.
>
> Despite the modest system size, we believe the experiments are sufficient to support the main claim of this work:
>
> > *joint TN-SS consistently discovers significantly more compact, and in some cases previously unknown, circuit structures compared with commonly used baselines.*
>
> This trend is observed in both the synthetic-operator experiments and the QFT benchmarks.
>
> To avoid any misunderstanding, we will further emphasize in the revision that the quantum results should be interpreted as _proof-of-concept validations of the framework_, rather than claims of large-scale quantum circuit synthesis. Scaling joint TN-SS to larger qubit systems is an exciting direction for future work.
>
> ---
>
> ### W2. *Optimization stability issues (e.g., with gradient/SVD-based methods) suggest robustness challenges in practice.*
>
> **R:**  Thank you for raising this point. The instability mentioned in the paper (also clearly stated in the Limitations section) arises from optimizing core tensors under a fixed TN structure, which is *not* the focus of this work. Joint TN-SS concerns model selection, whereas instability in core-tensor optimization is a *well-known open problem* in both tensor networks and quantum circuits, especially for network topologies containing loops.
>
> Therefore, we regard this instability as a *general limitation of the underlying optimization problem*, rather than a drawback of the proposed joint TN-SS framework.
>
> In practice, we mitigate these issues using standard cold- and warm-restart heuristics. As shown across all experiments, joint TN-SS remains stable and achieves strong results, despite the inherent challenges of core-tensor optimization.
>
> We will clarify this distinction more explicitly in the revised manuscript.

---

> > ### Author Response · Authors · 2025-11-18
> >
> > Dear Reviewer,
> >
> > Below, we continue with our responses to the remaining weaknesses raised in your review.
> >
> > ---
> >
> > ### W3. *“The significance to the broader ICLR community is not so clear as the method is specific to Tensor Network Operators.”*
> >
> > **R:**  Thank you for raising this point. We would like to clarify that tensor network operators (TNOs) are *parameter-efficient representations of linear operators*, and linear operators are _fundamental_ building blocks in almost all modern ML models, including transformers, MLPs, diffusion models, attention layers, and beyond.
> >
> > In recent years, tensor networks have become increasingly relevant in the ML community, appearing in: model compression [1,2],  LLM training and fine-tuning frameworks such as CoMERA, T-LoRA, and QuanTA [3–5],  reinforcement learning [6],   prompt learning [7],  statistical or high-dimensional estimation [8,9],  and physics-inspired / hybrid ML–physics models [10].
> >
> > These studies, together with foundational work on tensor decomposition and the existing TN-SS literature (including the TN-SS papers cited in our manuscript), show that tensor networks, and TNOs specifically, are already tightly connected to active research directions in the ICLR community.
> >
> > Our work contributes to this line by providing a *new, symmetry-guided framework* for structure search that can benefit a wide range of efficient-learning problems. In the revision, we will highlight these connections more clearly.
> >
> > **References**
> >
> > [1] Novikov et al., *Tensorizing Neural Networks*, NeurIPS’15.
> > [2] Kossaifi et al., *Tensor Regression Networks*, JMLR’20.
> > [3] Yang et al., *CoMERA*, NeurIPS’24.
> > [4] Tao et al., *T-LoRA*, ICCV’25.
> > [5] Chen et al., *QuanTA*, NeurIPS’24.
> > [6] Sozykin et al., *TTOpt*, NeurIPS’22.
> > [7] Qiu et al., *STEPS*, CVPR’25.
> > [8] Han et al., *Tensor Estimation Theory*, Ann. Stat.’22.
> > [9] Saiapin & Batselier, *TN Feature Learning*, AISTATS’25.
> > [10] Stoudenmire & Schwab, *Supervised Learning with Tensor Networks*, NeurIPS’16.
> >
> > ---
> >
> > ### W4. *“The novelty is limited as it borrows an elegant but relatively simple concept of symmetry breaking.”*
> >
> > **R:**  Thank you for your concern. We would like to use this opportunity to clarify the novelty of our contributions. Although the conceptual intuition from symmetry breaking is simple, the *technical novelty and scope of the contributions* are significant:
> >
> > - This is the *first* formulation of structure search for TNOs.  TNOs have recently emerged as an important tool for parameter-efficient learning, yet no prior work has studied how to automatically search their structures.
> >
> > - This is the *first* extension of standard TN-SS to the more practical and significantly more challenging _joint_ TN-SS setting.  Building on this formulation, we propose a simple and effective two-phase algorithm. The extensive experiments across three domains validate both the efficiency and performance advantages of the approach.
> >
> > - This is the *first* work that connects TN-SS to quantum circuit optimization.  Our results show that the proposed symmetry-guided structure search can automatically discover compact quantum circuit architectures. This is an application not explored before.
> >
> > We will emphasize these contributions more clearly in the revision.
> >
> > ---
> >
> > Thank you again for your constructive feedback. We welcome any further discussion.
> >
> > Best regards,
> > the authors

---

> > > ### Author Response · Authors · 2025-11-18
> > >
> > > Dear Reviewer,
> > >
> > > Thank you again for your constructive questions. Below, we respond to Q1–Q4. We hope the clarifications help address your concerns.
> > >
> > > ---
> > >
> > > ### Q1. *“How sensitive is joint TN-SS to the balance between the shared symmetry phase and the symmetry-breaking phase? Does tuning this trade-off require manual effort?”*
> > >
> > > **R:**  Thank you for the question. joint TN-SS is robust to phase transition.
> > >
> > > In our approach, the phase transition is determined entirely by the two task-transparent tolerances $(\eta_1, \eta_2)$.
> > >
> > > Empirically, joint TN-SS is *robust to a wide range of $(\eta_1, \eta_2)$*. For example, in Table 9, RSE remains less than $5\times{}10^{-10}$ for $\eta_2$ varies in 0.01, $10^{-4}$, $10^{-6}$, and $10^{-8}$ (and $\eta_1$ does not need to specificed explicitly in this experiment).
> > >
> > > Tuning is also simple: increase $\eta_1$ until the shared-structure loss reaches a reasonable tolerance, then choose $\eta_2 > \eta_1$ to allow mild task-specific variation.
> > >
> > > In short, the trade-off does not require heavy manual tuning, and the performance is stable across reasonable settings.
> > >
> > > ---
> > >
> > > ### Q2. *“Efficiency is mentioned many times—can you give quantitative estimates of the efficiency gains?”*
> > >
> > > **R:**  Thank you for the question. Yes. We provided quantitative estimates of the efficiency gains using the number of evaluation calls during the structure search. We report the evaluation cost in the main text or appendix.
> > >
> > > For example, in the tensor decomposition experiment , Figure 2 and Tables 5–8 clearly show that joint TN-SS achieves consistent reconstruction advantage using much fewer evaluations compared with conventional TN-SS. Figure 1(c) also visualizes this trend directly.
> > >
> > > We will add a short summary paragraph consolidating these quantitative efficiency gains in the revision.
> > >
> > > ---
> > >
> > > ### Q3. *“Have you explored alternative optimization schemes to improve robustness? Any ablation study?”*
> > >
> > > **R:**  Yes. For the inner minimization, we use different optimizers depending on the task. In tensor decomposition and PEFT, we use Adam, aligning with the baselines. For quantum experiments, we adopt an SVD-based optimizer to compute core tensors.
> > >
> > > We also tested gradient descent on Stiefel manifolds for the quantum case, but instability persisted  (thus we do not report it in the text), especially for network topologies with many loops. This phenomenon is consistent with well-known difficulties in both tensor networks and variational quantum circuits, and, as mentioned in our previous response, falls outside the main scope of this paper, which focuses on structure search rather than core-tensor optimization.
> > >
> > > ---
> > >
> > > ### Q4. *“How transferable are the discovered TN structures across related tasks?”*
> > >
> > > **R:**  Thank you for this interesting question. The current work focuses on discovering compact structures for specific tensors (e.g., tensor data, LLM weight matrices, or quantum operators). Under this setting, transferability is not the primary objective, and in our experiments we do not observe clear common structures that would suggest cross-task generalization of TN structures.
> > >
> > > A systematic study of structure transfer is indeed an interesting direction for future work. If you have a particular scenario in mind, we would be happy to discuss it in more detail.
> > >
> > >
> > > ---
> > >
> > > We appreciate your questions and time to read our responses. Please feel free to let us know if any point needs further clarification.
> > >
> > > Best regards,
> > > The authors

---

> ### Comment · Reviewer_TTpc · 2025-11-25
>
> Thank you for your polite replies
>
> W1. Can you please clarify/elaborate an what you mean by "resource constrains" in the limitations
>
> W2. I appreciate your thorough reply.
>
> W3. Can you please clarify how TNOs generalize to all TN networks? Is this work then general to TN or constrained to TNOs?
>
> W4. I appreciate the clarification, can you comment on the expected importance of this? Specifically, "discovering compact structures for the specific tensors" as you mention, the method is not transferable, so what is it immediate impact, or what is the outlook from this method? Is there something that we learn here that can be applied to other methods further down the line?
>
> Q1-4 Thank you for the clarifications.

---

> ### Author Response · Authors · 2025-11-26
> **Response and Revision to the Follow-up Question on W1/2**
>
> Dear Reviewer,
>
> Thank you very much for your thoughtful follow-up questions. In this post, we first address W1, W2 in detail. We hope these clarifications help resolve your concerns.
>
> ---
>
> ### W1. “Can you please clarify what you mean by ‘resource constraints’ in the limitations?”
>
> **R:**
> Thank you for giving us the opportunity to further clarify this point. By “resource constraints,” we refer specifically to the *runtime memory* limitations of our hardware that restrict the scale of the *quantum experiments* ( Appendix E.5 provides the hardware details).
>
> In the quantum experiments, we must load the full-dimensional target operator as “data”. The memory cost of these tensors grows *exponentially* with the number of qubits and typically requires a large-memory CPU server (implemented with C++ backends; you can find our code in https://anonymous.4open.science/r/test-927F). Because our available CPU environment had limited memory, we were unable to load and manipulate operators beyond a modest number of qubits.
>
> We understand that this limitation may raise concerns about scalability. To address this, we conducted extensive experiments in *multiple* domains, including tensor decomposition and PEFT, both implemented on GPUs. In these settings, we evaluated search efficiency across varying scales (Tables 6–8) and also tested on relatively large models (e.g., Llama2). These scales already exceed those studied in prior TN-SS work.
>
>
> Furthermore, Appendix C.6 provides a detailed computational analysis showing that the overall complexity per iteration is   $\mathcal{O}(LM)$, where $L$ is the number of core tensors and $M$ is the number of TNOs in the system. This analysis indicates that the method itself scales *low-order polynomially* in the structural parameters, suggesting potential for larger-scale quantum problems once adequate memory resources are available.
>
> ---
>
> ### “W2. I appreciate your thorough reply.”
> **R:** Thank you very much for the kind feedback.
>
> ---
>
> **Revision:**
> We have updated the main text to reflect the above clarifications and improve readability in following two points:
>
> 1. We revised the limitation statement for clarity: the phrase “resource constrains” is now explicitly written as “resource constraints ( Appendix E.5 provides the hardware details),” making the link to the implementation details in the appendix clearer and more precise. We also clarified that the observed scalability limitations mainly stem from the task-dependent inner optimization, which is not the focus of the current work.
>
> 2.  We expanded the first paragraph of Section 4.3 (quantum experiments) to better articulate the motivation, expected benefits, and current limitations of our quantum study.
>
> If any point remains unclear, we would be very happy to elaborate further.

---

> > ### Author Response · Authors · 2025-11-26
> > **Response and Revision to the Follow-up Question on W3**
> >
> > In this post, we continue to respond to the following-up questions on W3.
> >
> > ---
> >
> > ### W3. “Can you please clarify how TNOs generalize to all TN networks? Is this work general to TNs, or constrained to TNOs?”
> >
> > **R:**  Thank you for this important question. TNOs can generalize a very wide range of, but _not_ all, TNs.
> >
> > TNOs can represent a broad class of widely used TNs by appropriately reconnecting core tensors. Examples include MPS/MPO [1], isometric PEPS [2], Tree TNs (Hierarchical Tucker) [3], and MERA [4]. Moreover, with a sufficient number of core tensors, TNOs are known to universally approximate arbitrary linear operators (Barenco et al., 1995, Mottonen et al., 2024).
> >
> > However, not all TNs are covered in TNOs. TNOs represent a structured subset, specifically, the TNs whose connectivity can be expressed in a “circuit-like” graph. In this sense, our work is complementary to the existing TN-SS approaches: prior TN-SS may search general network topologies but restrict the number of core tensors to match the tensor modes, whereas TNOs allow an arbitrary number of cores but within a constrained circuit-style structure.
> >
> > For your second question, our work focuses on TNOs not general TNs, with the following three reasons: 1) TNOs offer strong practical benefits in recent ML studies (Chen et al., 2024a, Li et al., 2025); 2) they align naturally with quantum circuit representations and quantum ML; 3) structure search over TNOs is technically unexplored and challenging due to the additional structural constraints. We believe the TNO restriction would make TN-SS discover novel and *practically useful* TN models.
> >
> > ---
> >
> > **Revision:**
> > We added a detailed paragraph in Appendix B (due to the page limit) to clarify the representational boundary between TNOs and general TNs. We also added a pointer in the main text (in the TNO definition paragraph) to guide readers directly to this explanation.
> >
> > ---
> >
> > **Additional References:**
> > [1] Martin, A., et al. “Combining matrix product states and noisy quantum computers for quantum simulation.” *Phys. Rev. A* 109.6 (2024): 062437.
> > [2] Slattery, L., & Clark, B. K. “Quantum circuits for two-dimensional isometric tensor networks.” arXiv:2108.02792 (2021).
> > [3] Seitz, P., et al. “Simulating quantum circuits using tree tensor networks.” *Quantum* 7 (2023): 964.
> > [4] Luchnikov, I. A., et al. “Simulating quantum circuits using the multi-scale entanglement renormalization ansatz.” arXiv:2112.14046 (2021).

---

> > > ### Author Response · Authors · 2025-11-26
> > > **Response and Revision to the Follow-up Question on W4**
> > >
> > > ### W4. “What is the immediate impact of this method if it is not transferable? What is the outlook? Is there something we learn that can be applied elsewhere?”
> > >
> > > **R:**  Thank you for these deep questions. We clarify both the immediate impact and the broader outlook.
> > >
> > > Practically, our method can be used immediately to strength the express power of TNs as parameter-efficient representation in ML. Whether for low-rank model compression, PEFT or related topics, our method provides a systematical and more efficient framework to guide practitioner to solve the challenging model selection issue when using tensors.
> > >
> > > For quantum circuits, the results support our medium-term goal that (joint-)TN-SS can naturally contribute to practical quantum problems, not just quantum-inspired ML. In the current NISQ era, a more compact circuit directly means less latency, less noise accumulation, and a more stable physical implementation. This is a concrete benefit for near-term quantum computing.
> > >
> > > Conceptually, our work bridges ML and computation with classical physical principle. we believe that the  “symmetry-breaking” perspecitve carries general insights. As highlighted in Lines 214–215:
> > >
> > > > *“The optimal solution to a difficult problem often lies close to a suboptimal solution found in a coarser, lower-dimensional search space”.*
> > >
> > > This principle is not limited to tensor networks. It has the  potential to inspire future developments in neural architecture search, high-dimensional combinatorial optimization and other discrete problems where a coarse-to-fine search strategy can dramatically improve efficiency.
> > >
> > > ---
> > >
> > > **Revision:**
> > > We added a new paragraph in the concluding remarks to highlight these potential impacts and outline promising future directions (The limitation part is moved to Appendix A due to the page limit).
> > >
> > > ---
> > >
> > > Thank you again for your thoughtful questions and for engaging deeply with our work. Your feedback helped us further improve the manuscript. We welcome any further discussion or suggestions.
> > >
> > > Best regards,
> > > The authors

---

### Official Review · Reviewer_quGS · 2025-11-01

**Soundness:** 3
**Presentation:** 3
**Contribution:** 3
**Rating:** 6
**Confidence:** 4

**Summary:**

The authors provide a Heuristic method for connected tensor operators structure search algorithm, from the intuition of the symmetry breaking. The algorithm provide a efficient way to optimize the tensor structure will considering multiple operators together, and have good performance considering the simplicity.

**Strengths:**

1. The first paper which have a practiclly useful algorithm for tensor structure search for multi-operator system
2. demonstrate application in LLM finetuning and operator optimization in quantum circuits.

**Weaknesses:**

1. The narrative need improvement.For example, the author demonstrated the second phases's optimization strategy, while the frist phase, hwo to proposed new G, and the graph optimization procedure is lacking.

**Questions:**

1. How scalable is this algorithm? What is the complexity in each part of the optimization?

2. For circuit optimization, is that applicable to all kind of quantum operators?

---

> ### Author Response · Authors · 2025-11-15
>
> Dear Reviewer,
>
> We sincerely thank you for your constructive feedback and insightful questions. Below, we address the concerns raised in the *Weaknesses* and *Questions* sections, and clarify how we will further improve the paper.
>
> ---
>
> ### 1. *“The narrative needs improvement. For example, the author demonstrated the second phase's optimization strategy, while the first phase—how to propose new $G$ and the graph optimization procedure—is lacking.”*
>
> **R:** Thank you for highlighting this important point. In our work, Phase I directly coincides with existing TN-SS algorithms, as described in Section 4.1. For this reason, we kept the introduction of Phase I in the main text concise.
> However, to make the paper self-contained, we have already reintroduced the full procedures of Phase I in Appendix A.3. Since the goal of Phase I is to identify a shared graph $G$, this phase can directly leverage standard TN-SS methods (e.g., TNGA, TNLS, TnALE, Greedy) with only minor adaptations required for the TNO setting.
>
> In the revision, we will explicitly emphasize the workflow and strategy of Phase I in the main text, clearly referencing the TN-SS algorithms and describing their roles and necessary modifications for TNOs. We appreciate your suggestion. This will indeed help improve the clarity and completeness of the narrative.
>
> ---
>
> ### Questions
>
> ### 1. *“How scalable is this algorithm? What is the complexity in each part of the optimization?”*
>
> **R:**  Thank you for raising this crucial issue. We summarize the scalability and computational complexity of our method below:
>
> - **Scalability:**
>  The method scales well across a wide range of structure-related parameters $(Q, I, M)$, as demonstrated empirically in Tables 6–8 of Appendix A.6.
>
> - **Complexity:**
>   The computational complexity depends on the TN-SS algorithm used. For example, when using TnALE in Phase I, the typical cost for the whole joint TN-SS is $\mathcal{O}(LM)$ per iteration, where $L$ denotes the number of core tensors and $M$ denotes the number of TNOs involved in the optimization. If a different TN-SS algorithm is used, additional factors naturally arise, such as the population size in TNGA or the number of sampled structures in TNLS. The proposed method is scalable from the structure search perspective.
>
> - **Meta-algorithm nature of (Joint) TN-SS:**
>   It is also important to note that joint TN-SS is a “meta-algorithm”. Its total complexity additionally depends on the inner minimization in Eq. (5), which varies by task (tensor decomposition, PEFT, quantum circuit optimization). This bi-level optimization nature inherently influences the computational difficulty.
>
> In the revision, we will include a clear summary of the computational costs for each phase, together with the empirical scalability results.
>
> ---
>
> ### 2. *“For circuit optimization, is this applicable to all kinds of quantum operators?”*
>
> **R:**  Yes. The proposed method is applicable to *all* types of quantum operators. The TNO representation is expressive enough to encode any quantum operator, and our joint TN-SS framework operates directly on TNOs without assuming any special structure. Therefore, the method naturally handles arbitrary quantum operators within the optimization process.
>
> We will make this point explicit in the revised manuscript to avoid ambiguity.
>
> ---
>
> We greatly appreciate your thoughtful comments and suggestions. The clarifications above, together with the planned improvements in narrative and technical detail, will help strengthen the clarity and rigor of our paper. We welcome any further discussion and feedback.
>
> Best regards,

---

### Author Response · Authors · 2025-11-30
**Summary of Reviewer Feedback and Author Revisions**

Dear Reviewers and AC,

Thank you all for your dedicated work in reviewing our paper during this busy and turbulent period. We truly appreciate your time:). Below is a brief summary of the reviewers’ feedback, along with our responses and the revision record during the discussion phase.

---

### **Score summarization:**

Among five reviews, three reviewers (AEK4, ADcS, quGS) gave clear positive or acceptance-leaning recommendations. One reviewer (TTpc) provided a borderline but constructive review with several follow-up questions, and one reviewer (rZHz) also gave a borderline score while *explicitly* noting limited expertise in this area. No reviewer requested major further changes.

---

### **Positive comments by reviewers:**

Reviewer *quGS* highlighted that this is the ***first practically useful algorithm*** for tensor structure search in multi-operator systems, with ***strong applications*** demonstrated in LLM fine-tuning and quantum circuits.

Reviewer *TTpc* also praised the ***novel*** symmetry-based perspective, the ***versatility across domains***, and the ***clear limitations*** and related work, noting that the method “balances shared efficiency and individualized expressivity.”

Reviewer *AEK4* emphasized the ***novel*** extension from single-operator to multi-operator TN-SS, the clear method design, the ***broad and compelling empirical results***, and the efficiency gains from the shared-then-specialized search strategy.

Reviewer *rZHz* praised the well-defined problem setup, the straightforward and general approach, the ***interpretability of the tolerance-based design***, and the comprehensive experimental coverage.

Reviewer *ADcS* noted that the results are ***consistently strong across settings***, the revised paper is now much more accessible, and ultimately recommended acceptance, stating that the contribution is solid and relevant to a ***meaningful subset of the ML community***.

---

### **Addressed concerns:**
We have carefully addressed *all* substantive reviewer concerns through detailed clarifications, concrete revisions, and additional numerical experiments.

For Reviewer [*quGS*, original score: 6], the concern is mainly on the clarity of Phase I of our method. We addressed this by explicitly pointing to the full procedure in Appendix A.3 and refining the main text to emphasize that our method is plug-and-play. Phase I directly relies on existing TN-SS algorithms, and its details were omitted earlier only to avoid unnecessary repetition.

For Reviewer [*TTpc*, original score: 4], we clarified the scalability and staiblity mentioned in the limitation part, explained that ***scalability and stability issues arise from task-dependent inner optimization (not the focus of this paper)***, strengthened the case for broader ML relevance, and highlighted both conceptual and practical impact.
*The reviewer explicitly acknowledged our reply with the stability clarification (“W2. I appreciate your thorough reply.”)* and then asked three follow-up questions that we then fully answered before OR disabled further reviewer responses.

For Reviewer [*AEK4*, original score: 8], we clarified the scalability limits of the quantum experiments, explained why Phase-II is a consistent approached verified by experiments, after which the reviewer stated all concerns were resolved and reaffirmed a ***strong*** acceptance recommendation.

For Reviewer [*rZHz*, original score: 4], we provided clearer guidance showing that our extensive numerical results provide ***sufficient evidence to support the robustness*** of Phase II, clarified the PEFT and quantum experiment scope, released anonymous quantum-code for reproducibility, and gave a more intuitive explanation of the theory. The reviewer explicitly expressed rather limited domain familiarity (*“ I want to remind the AC that this is not my research field and I have rather limited knowledge on it, so please take it with a grain of salt.”*) and maintained a neutral stance.

Finally, for Reviewer [*ADcS*, original score: 4 → acceptance before the explosion of the info leak], we improved accessibility by revising the introduction and appendix, adding intuitive einsum examples, refining the physics metaphor, clarifying TN relevance to ML, and explaining the PEFT scope, leading to a raised score and ***a clear acceptance recommendation***.

---

### **Revision record:** (highlighted in GREEN)

30 Nov.  -- Added new PEFT results in Table 1 with four additional baselines.

26 Nov. -- Clarified broader ML impact and added clearer links to Appendix sections.

17 Nov. -- Added new references and an intuitive introduction to joint TN-SS in Appendix A and strengthened discussions on scalability, stability, and ML relevance.

---

We thank all reviewers and AC for your engagement, especially during the challenges by the OR incident. We appreciate your fairness and professionalism, and hope our responses address all concerns clearly.

Best regards,

---

### Meta-Review · Area_Chair_bsYa · 2026-01-04

**Summary:**

This is a borderline paper with scores (4,4,4,6,8) that proposes an extension to tensor networks.

Some of the concerns include issues with the paper's presentation, the potentially unstable nature of the symmetry-breaking phase of the algorithm, the extent to which the approach is novel, the strength and scale of the empirical evidence, and the broader relevance for the ML community.

The authors provide a correct summary of the discussion, emphasizing the positive aspects of the paper.

**Reviewer Concerns:**

The authors gave a pretty good summary of the positive aspects in the discussion.
Some negative aspects that were not present in the summary are the scalability issues brought up by reviewer TTpc, which seem to be an actual limitation of the approach. This is why the experimental results were presented in small examples. That's an issue that several reviewers brought up.
Reviewer AEK4 was satisfied with the responses and recommended to accept the paper.
Reviewer rZHz, was not completely satisfied with the answers provided by the authors.
Reviewer ADcS was not satisfied with the paper's presentation and explanations. I agree with this reviewer that the paper could improve how it provides intuition and motivation for the approach.

**Reviewer Scores:**

One reviewer raised their score from 4 to 6. I don't know if the other reviewers would have changed their scores.

---

### Decision · Program_Chairs · 2026-01-26

Reject